# Wind-driven stratification patterns and dissolved oxygen depletion off the Changjiang (Yangtze) Estuary

Taavi Liblik[1,2], Yijing Wu[1], Daidu Fan[1,3,*],Dinghui Shang[1]

[1] State Key Laboratory of Marine Geology, Tongji University, Shanghai, 200092, China
[2] Department of Marine Systems, Tallinn University of Technology, Tallinn, 12618, Estonia
[3] Laboratory of Marine Geology, Qingdao National Laboratory for Marine Science and Technology, 387P+WW Shinan, Qingdao, Shandong, China

*Correspondence to*: Daidu Fan (ddfan@tongji.edu.cn)

**Abstract.** The area off the Changjiang Estuary is under strong impact of fresh water and anthropogenic nutrient load from the Changjiang River. The seasonal hypoxia in the area has variable location and range, but the decadal trend reveals expansion and intensification of the dissolved oxygen (DO) depletion.

Two oceanographic cruises, conducted in summers 2015 and 2017, were complemented by river discharge, circulation simulation, and remotely sensed wind, salinity and sea level anomaly data. Two distinct situations of stratification and DO distributions in summers 2015 and 2017 can be explained by wind forcing and concurrent features in surface and deep layer circulation, upwelling and downwelling events.

Enhanced primary production in the upper layer in the Changjiang Diluted Water (CDW) or in the upwelled water determined the location and extent of oxygen depletion. However, pycnocline created by Kuroshio subsurface water intrusion was essential precondition for hypoxia formation. Spreading of the CDW, occurrence of upwelling and intrusion of deep layer water are strongly influenced by wind forcing.

Intensification of Chinese Coastal Current (CCC) and CDW spreading to south together with coastal downwelling caused by northerly wind were observed in 2015. This physical forcing lead to well ventilated area in north and hypoxic area of $1.3 \times 10^4$ km$^2$ in the southern part. Alteration of CCC due to Ekman surface flow and reversing the geostrophic current related to upwelling induced by summer monsoon (southerly winds) were observed in 2017. Wind-driven offshore advection in the surface layer caused intrusion of Kuroshio derived water to shallow (<10 m depth) areas at coastal slope. Intense hypoxia (DO down to 0.6 mg l$^{-1}$) starting from 4-8 m depth connected to CDW and deep water intrusion in north, and coastal hypoxia linked to upwelling were observed in 2017.

Variability in wind forcing and river run-off and related changes in physical environment considerably shape extent and location of hypoxia in synoptic, seasonal and inter-annual time-scale off the Changjiang Estuary.

## 1      Introduction

Dead zones in the coastal ocean have spread since the 1960s (Diaz and Rosenberg, 2008). Besides eutrophication (Diaz and Rosenberg, 2008), climate change (Altieri and Gedan, 2015) intensifies dissolved oxygen (DO) depletion. Estuaries and other regions of fresh water influence are typically very productive regions and often experience hypoxia (Conley et al., 2009; Cui et al., 2019; Obenour et al., 2013; Testa and Kemp, 2014). Natural

and anthropogenic nutrient load in these areas leads to intensive sedimentation of organic matter. DO is used by concurrent decomposition of detritus by bacteria in the sub-surface layer. Deep layers below the euphotic zone can receive DO by physical processes only – lateral advection and vertical mixing. If the decline of DO exceeds DO import or production, the DO concentrations decrease.

Hypoxia is a natural phenomena off the Changjiang Estuary, as evidences of its existence extend back to 2600

60       years at least (Ren et al., 2019). However, hypoxic area has been expanded in recent decades (Chen et al., 2017; Ning et al., 2011; Zhu et al., 2011), which can be mostly related to eutrophication (Wang et al., 2016). A low DO zone off the estuary can be found from spring (Zuo et al., 2019) until decay of stratification in autumn (Wang et al., 2012). The formation and maintenance of seasonal hypoxia has been related to various physical, biogeochemical and biological processes (Ning et al., 2011). Hypoxia off the Changjiang Estuary is in literature

often divided to the southern and northern areas (Wei et al., 2017a; Zhu et al., 2011). The northern part features shallow (20 – 40 m) and flat sea bottom while the southern part is characterized by a deep trough with water depth > 60 m and a steep slope.  Numerous small peninsulas and islands occupy the southern part with irregular coastline.

The area is intensely impacted by huge river discharge from the Changjiang (Beardsley et al., 1985; Xu et al.,

2018). Correlation between the CDW plume area and river discharge has been suggested by Kang et al. (2013). Nutrient rich freshwater mixes with ocean water to form the Changjiang Diluted Water (CDW). Enhanced primary production in the CDW causes intense detritus accumulation and DO consumption in the near-bottom layers (Chen et al., 2017; Li et al., 2018b; Wang et al., 2017, 2016; Zhou et al., 2019). The CDW is separated from the deeper water with a shallow halocline, which often coincides with the seasonal thermocline (Zhu et al., 2016). The shallow

pycnocline impedes vertical mixing and vertical DO transport downwards. Thus, the presence of the CDW provides favorable conditions for hypoxia formation in the layer below the pycnocline. Moreover, the shallow pycnocline supports accelerated warming in summer (Moon et al., 2019) and as consequence the pycnocline strengthens even more. Therefore, the spreading of the CDW strongly determines oxygen consumption and depletion in the near bottom layer.

Shallower areas near the river mouth are strongly impacted by tidal forcing (Li et al., 2018). However, in the open sea the contribution of tides in the vertical mixing budget is low comparing to the wind stirring (Ni et al., 2016).

Abrupt wind mixing events can weaken stratification and considerably increase DO concentrations in the deep layer (Ni et al., 2016). However, such vertical mixing also cause nutrient flux to the upper layer (Hung et al., 2013) and as a result primary production in the upper layer and DO consumption in the deep layer are enhanced (Ni et al., 2016).

Circulation and hydrography near the Changjiang Estuary are besides the CDW and wind mixing impacted by the Chinese Coastal Current (CCC), and the Taiwan Warm Current (TWC) originates from the Taiwan Strait warm water in the surface and the shelf-intrusion water of Kuroshio subsurface water (KSSW) in the bottom (Lie and Cho, 2002, 2016; Liu et al., 2017; Zhang et al., 2007b; Zuo et al., 2019). Geostrophic currents in the area are modified by wind forcing, which exhibits seasonal alternation of winter and summer monsoons (Lie and Cho, 2002). Winds from north and northeast prevail during the winter monsoon and southerly winds dominate during the summer monsoon (Chu et al., 2005). Summer monsoon favors upwelling formation in the East China Sea (ECS) (Hu and Wang, 2016), including sea areas nearby the Changjiang Estuary (Xu et al., 2017; Yang et al., 2019). Upwelling events bring large amounts of nutrients to the surface layer (Wang and Wang, 2007), which cause increase of primary production and decrease of DO concentrations in the near-bottom layer (Chen et al., 2004; Zhou et al., 2019). Moreover, coupling of the CDW plume and KSSW upwelling has a cumulative effect on primary production and hypoxia formation in the near bottom layer (Wei et al., 2017a). CDW spreading and subsurface water upwelling are both sensitive to wind forcing, which gives rise of the hypothesis: hypoxic conditions are sensitive to wind.

Recent modeling study (Zhang et al., 2018) has demonstrated that the CDW is very mobile and as a result location and areal extent of bottom hypoxia is variable as well. Thus, distributions of oceanographic fields mapped during an occasional research cruise could depend on simultaneous synoptic scale forcing conditions and does not necessarily represent the hydrographical and chemical situation in a particular summer. In order to put oceanographic cruise results in more general context, observed ocean variables must be linked to the forcing. The role of nutrient consumption and primary production in the CDW, in the upwelled water and their interaction have been investigated. It is clear that enhanced primary production due to upwelling and in the CDW results higher oxygen consumption rates in the near bottom layers (Chen et al., 2017, 2004; Große et al., 2019; Li et al., 2018b; Wang et al., 2017, 2016; Wei et al., 2017a; Zhou et al., 2019). The focus of this work will be on the physical forcing mechanisms that create favorable conditions for the aforementioned biogeochemical and biological processes.

The hypothesis of the present study is that prevailing physical factors, wind forcing and river run-off mainly determine the stratification patterns and as consequence the location and areal extent of the DO depletion off the Changjiang Estuary. This means that changes in forcing over various time scales (synoptic, inter-annual, decadal) also cause changes in the patterns (e.g., the stratification or the location of the hypoxic area). To testify this hypothesis, we (1) describe the observed patterns (horizontal and vertical distributions) of hydrography, chlorophyll *a* and DO off the Changjiang Estuary during the two cruises in the summers of 2015 and 2017, (2) explore the potential link between physical variables and DO distributions, and (3) investigate the forcing mechanism behind observed patterns.

**2 Data and Methods**

We have performed two surveys in the Changjiang Estuary and the adjacent sea on board the RV Zhehaike-1 from 27 August to 5 September in 2015 and from 24 to 29 July in 2017, respectively. CTD (Conductivity, Temperature, Depth) and DO profiles were obtained from 110 stations in 2015 and 83 stations in 2017. However, we have excluded stations inside river mouth, therefore 65 and 49 stations were included in the present study, respectively in 2015 and 2017 (Fig. 1). At all stations vertical profiles of temperature, salinity and DO were recorded with the

SBE 25plus Sealogger CTD with DO sensor SBE43. The DO sensor was calibrated against water sample analyses conducted by Winkler titration method. Linear regression between sensor and sample DO data for 2015 and 2017 were analyzed: $DO = DO_{SBE} \times 0.98 – 0.09$ ($r^2 = 0.93$ $n = 191$) and $DO = DO_{SBE} \times 0.98 – 0.06$ ($r^2 = 0.99$ $n = 98$), where $DO_{SBE}$ is the DO recorded by SBE43 sensor (Wu et al., 2019).

For the 2015 cruise, chlorophyll a (Chl a) concentrations were measured from water samples as follows. About

135 500ml surface water was filtered through 0.7μm (pore size) Whatman glass fiber filters at every station with pressure lower than 15Kpa to get the particles for further analysis. To avoid the influence of light during the storage, all filters after filtration were fold up and wrapped by the aluminum foil before being put in the -20℃ freezer. The pigments in each filter were extracted by 90% acetone and measured by fluorometric analysis using the Hitachi Designs fluorescence spectrophotometer (F-2700).

Daily satellite (Aqua MODIS) derived Chl *a* data were downloaded from https://coastwatch.pfeg.noaa.gov/. According to data availability mean Chl *a* field from 21-22 July, i.e. 3-4 days before our survey, is presented in the paper to illustrate primary production in 2017.

Salinity data in the present work is given in the Practical Salinity Scale (Fofonoff and Millard Jr, 1983) and density as potential density anomaly ($\sigma_0$) to reference pressure of 0 dbar (Association for the Physical Sciences of the Sea,

2010).

8-day running average Sea Surface Salinity (SSS) with resolution of 0.25° calculated from Soil Moisture Active Passive (SMAP) mission (Meissner et al., 2018) was used for the spatial extent estimation of the CDW from 2015 to 2018. Remotely sensed salinity well agreed with the in-situ salinity measurements (Fig. 2) at the mooring station M1 (Fig. 1).

Reprocessed 6 hourly wind observations with horizontal resolution of 0.25° from 1993 to 2018 downloaded from Copernicus Marine Service (product ID WIND_GLO_WIND_L4_REP_OBSERVATIONS_012_006) was used to describe wind conditions. Wind stress was calculated following Large and Pond, (1981):

$$\tau = \rho_{air}\, C_D\, |U|U_{10}, \tag{1}$$

where $C_D$ is the drag coefficient and $U_{10}$ is the wind speed at the reference height of 10 m above sea level.

Current speed and direction from the Copernicus Marine Service reanalysis product GLORYS12V1 from 1993 to 2018 was downloaded. The global eddy resolving model has regular horizontal grid with approximately 8 km (1/12°) resolution and standard vertical levels with resolution increasing from 1 m near the surface to 8 m around 50 m depth and 17 m around 100 m depth. Remotely sensed and in-situ observed temperature, salinity, and sea

level were assimilated to the model. The model has too rough resolution to estimate details of meso- and finer-scale features in the area. However, it can be used to estimate the general current patterns.

Daily mean sea level anomaly and meridional and zonal components of absolute geostrophic velocity based on satellite altimetry were downloaded from the Copernicus Marine Service (product DATASET-DUACS-REP-GLOBAL-MERGED-ALLSAT-PHY-L4). The sea level anomaly is defined as the sea surface height above the mean sea surface during reference period 1993-2012. Horizontal and temporal resolution of the dataset are 0.25° and one day, respectively.

Depending on marine organisms, hypoxia can have various definitions (Vaquer-Sunyer and Duarte, 2008). In the present study, hypoxia was defined as DO concentration of $<3.0$ mg $l^{-1}$. In order to estimate oxygen consumption, the apparent oxygen utilization (AOU) was estimated. AOU is the difference between DO concentration at the saturation level and measured DO concentration in water with the same temperature and salinity and is calculated as follows

$$AOU = DO_s - DO_M; \tag{2}$$

where $DO_S$ is DO concentration at the saturation level (Weiss, 1970) and $DO_M$ is the measured DO concentration. Total AOU (g $m^{-2}$) in the water column was calculated as

$$AOU_{TOT} = \sum_{z=0}^{z=h} \begin{array}{l} 0, if\ AOU \leq 0 \\ AOU(z)dz, if\ AOU > 0 \end{array} \tag{3}$$

where AOU(z) is the AOU profile and $h$ is the water column depth.

The upper boundary of the DO depletion (UBD) was defined as the AOU isoline 2 mg $l^{-1}$. Different authors have used various criteria to define water masses in the region. The cold bottom water in the area mostly origins from the KSSW (Liu et al., 2017). Therefore we define deep layer water as the upper boundary of the Kuroshio (UBK) subsurface water mass as isotherm 24.5 °C (Zhang et al., 2007a).

Daily (2015 and 2017 summer) and monthly (2001-2018) river discharge data from Datong hydrological station (see location in (Xu et al., 2018) were used in analysis.

Width of the buoyant coastal current was estimated according to (Lentz and Helfrich, 2002) as

$$W_p = (c_w/f)\ (1+c_w/c_a), \tag{4}$$

where $c_w = (g'\ h_p)^{1/2}$; $c_a = \alpha g'/f$; $h_p = (2Qf/g')^{\frac{1}{2}}$; $g' = gd\rho/\rho$ \hfill (5)

where f is Coriolis force, Q is the volume transport (taken as river discharge), $d\rho$ is the density difference between the plume and ambient fluids ($\rho$) and $\alpha$ is the coastal slope.

## 3 Results

### 3.1. Spatial patterns of hydrography, Chl *a* and oxygen

In order to link the thermohaline structure and DO observations we next analyze temperature, salinity, stratification and DO distribution patterns observed in summers 2015 and 2017. In order to illustrate primary production, subsequent organic matter degradation and oxygen consumption Chl *a* data is presented.

The water of SSS < 30 was observed along the coastal zone of the entire study area in 2015 (Fig. 3a). The 25 isohaline reached to 29.3° N in the southern part of the study area while its northern boundary was at 31.7° N near the mouth of the Changjiang Estuary in 2015. In 2017 the 25 isohaline reached only to 30.6° N in the southern part of the study area (Fig. 3b). While fresher water than 25 was also observed in the east and northeast off the Changjiang Estuary. Thus, the CDW southward spreading has been prevailing before the survey in 2015, but the CDW eastern and northeastern diversion was observed prior to survey in 2017.

The surface temperature (SST) was generally lower in the study area in 2015 (Fig. 3c). The maximum SST in the study area was 28.3 °C (31.3 °C) in 2015 (2017). Colder surface water was observed along the coast between 28.5° N and 31° N, and in the northern part of the study area (>31.5° N) in both years (Fig. 3c-d). The SST was much lower in these regions, but colder area covered smaller areas in 2017 (Fig. 3d) comparing to 2015.

The near-bottom temperature difference between the two surveys was particularly large in the northern part of the study area (Figs. 3g-h, 4a, 5a). Near-bottom temperature was 24-25 °C at 20-30 m sea depth in 2015 (Fig. 4a) while it was 21-22 °C in 2017 (Fig. 5a). Likewise, near-bottom water was saltier in 2017 (Figs. 3e-f, 4d, 5d). The location of minimum near-bottom temperature was at 50-60 m sea depth in the southern and central part of the study area in 2015 (Fig. 3g and Fig. 4b-c). The minimum temperature zone was closer to the coast and at shallower depths in 2017 (Fig. 3h and Fig. 5a-c).

Spatial patterns of stratification in the study area are described by the difference between the surface and near-bottom layer density (Fig. 6a-b). Weaker stratification (< 3 kg m$^{-3}$) was observed in the northern part of the study area in 2015 (Fig. 6a). KSSW was present only at the offshore part in north (Figs. 4a,d). Stratification was stronger (>3 kg m$^{-3}$) in the area south off the Changjiang Estuary, being particularly strong (>6 kg m$^{-3}$) in areas of 20-30 m bottom depth (Fig. 6a). In this area, the CDW covered the surface layer (Figs. 4e-f), and cold and saltier water mass occupied the near-bottom layer (Fig. 4b-c, e-f). Near the coast and offshore such strong stratification was not observed (Fig. 6a), i.e., the dense deep layer water did not intrude into the shallower areas and the CDW did not spread further offshore (Fig. 4a-c and Fig. 3 a,c,e,g). The UBK had considerable inclination from 24 m depth near the coast to 48 m depth at 80-85 km distance across shore in the southern section in 2015 (Figs. 4c,f and 7c). Opposite inclination, i.e. downwelling, was observed in the central station in 2015 (Fig. 4b,e). The boundary was at 38 m depth at the base of coastal slope, but at 17 m depth further offshore (Figs. 4b,e). Downwelling had pressed the salinity 30 isoline down to 40 m depth in the central section in 2015 (Fig. 4e). Probably onshore advection of more saline offshore water was pushed over fresher coastal water and caused vertical mixing.

Strongest stratification (>6 kg m$^{-3}$) was observed in the northern part of the study area in 2017 (Fig. 6b). The cold UBK was located at depths of 4-8 m in most of the northern section (Figs. 5a, 7d). Above the strong pycnocline was warm and fresh water, the CDW (Fig. 5a,d). The areal extent of very strongly stratified (>6 kg m$^{-3}$) region fitted with the area of CDW distribution in the surface layer (Fig. 3b). Stratification was weaker (<3 kg m$^{-3}$) in the very shallow areas (< 10 m, Fig. 6b), where cold and saltier water mass did not exist and in the southeastern part of the study area, where the CDW did not spread and bottom water was warmer than at the core of the cold water mass (Figs. 3f,h and 5a-f). Stratification pattern in the central and southern section was characterized by strong inclination of the isopycnals. The UBK was at 50 m depth at 110 km distance (from the western end of the section) offshore and at 7-8 m near the coast (0 km distance) in the southern section (Fig. 5c,f). Surfaced UBK, i.e. upwelling was observed at 70-80 km distance in the central section (Fig. 5b). Warmer and fresher surface water

were observed both shoreward and offshore side from the upwelling core (Fig. 5b, e). Thus, coupling of the riverine fresher water and upwelling could be expected there.

Low DO concentrations (2-3 mg l$^{-1}$) in the near-bottom layer occurred in areas of 25-60 m bottom depth in the southern and central part of the study area in 2015, affected an area of $1.3 \times 10^4$ km$^2$ approximately (Figs. 4h-i, 6e). Higher Chl $a$ values along the coastline in the central and southern part illustrate the role of primary production in the CDW and subsequent processes resulting oxygen consumption in the deeper layers (Fig. 6c). Although due to downwelling the low DO water mass is pressed deeper and further offshore while CDW is constrained closer to the coast (Figs. 4e h). Higher Chl $a$ value in one station in the northern part (at 31.6° N and 122.6° E) coincided with the lower (<30) surface salinity intrusion (Fig. 6a). Elevated Chl $a$ values further offshore (around 31.5° N and 123.2-123.5° E) did not coincide with the CDW (Figs. 3b, 6c). UBD and UBK were strongly linked in the southern and central section in 2015 (Figs. 4j-k, 7a,c). However, the boundaries coincided only in the offshore side at the northern section (Fig. 4j). This was the only section shown in Figs. 4-5, where the UBK and UBD did not coincide (Fig. 4j). Likewise, this region differs from the rest (except shallower areas with sea depth <20 m) of the observations by high near bottom DO values (Figs. 4g, 6e). The across-shore lateral distribution of DO, AOU and temperature under the thermocline reveal the linked structure in the northern section in 2015 (Figs. 4a,g,j). The coldest and lowest DO water is located in the offshore side of the section. Around 40-50 km distance, there is lateral maximum of the DO and temperature. Slightly lower DO and temperature were observed at 10-30 km. Similar lateral alternation from higher to lower and back to higher Chl $a$ (Fig. 3c) and in opposite phase in the surface temperature and surface AOU (lower-higher-lower, Fig. 4a,g), respectively, can be found in the northern section in 2015 (Fig. 3c). We discuss about possible reasons of such distributions in next subchapter. Near-bottom DO concentrations further offshore (east) and in the north were 3-5 mg l$^{-1}$ and higher in shallower depths in 2015 (Figs. 6e, 4g-i). Low DO area (2-3 mg l$^{-1}$) only partly overlapped with the CDW distribution, but fitted well with the area occupied by cold (< 20 °C) water mass in the near-bottom layer (Figs. 4a-c, 4g-l). The vertically integrated AOU maximum (>150 g m$^2$) pattern was observed between 30-31° N and 123-124° E in 2015 (Fig. 6g). Latter area did not coincide with the Chl $a$ distributions. In fact very low Chl $a$ values were observed at 30.5-31° N and 123-124° E (Fig. 6c) while oxygen depletion (the total AOU) was there rather high (Fig. 6e). Latter indicates that oxygen depletion in the southern and central part has developed over longer periods.

Two low DO concentration zones were observed in the near-bottom layer in 2017 (Figs. 5g-i and 6f). The one in the north well overlapped with the low salinity CDW and strong stratification area (Fig. 6b, 6f). Hypoxia there started at 4-8 m depth already and was closely linked to the location of the pycnocline (Figs. 5a, 5d, 5g). Lowest DO and highest AOU of all our measurements were observed in the northern part in 2017 (Figs. 5g,j). Some profiles showed DO nearly 0.6 mg l$^{-1}$ and AOU over 6 mg l$^{-1}$. High Chl $a$ were observed in the northern part, displaying the intense production in the upper layer (Fig. 6d). The second low DO concentration zone was observed in shallow depths (10 - 30 m) at the coastal slope in the southern and central part of the study area and overlapped with lower SST and SSS upwelling region (Figs. 5b-c, e-f, h-i). This indicates that coupling between coastal upwelling, which bring subsurface water to shallower depths (euphotic zone) and fresher riverine surface water, created favorable conditions for the primary production in the upper layer and concurrent DO consumption and depletion in the deep layer (Figs. 5h,i,k,l). Higher Chl $a$ concentrations in the upwelling areas revealed from the satellite image, although not as high as in north (Fig. 6d). The vertically integrated AOU had highest values (>150 g m$^2$) in the northern part of the study area in 2017 (Fig. 6h).

The high AOU zones in the near-bottom layer can be found in regions of different depths, e.g., as shallow as 4 m in 2017 (Figs. 4j-l, 5j-l). Strong and shallow enough stratification makes DO depletion possible in such a shallow depth. The question is whether the DO depletion is linked to a certain physical property of water, e.g. to a temperature-, salinity- or density isoline. We found high correlation ($r^2 = 0.81$, $p < 10^{-10}$, $n = 44$ in 2015, and $r^2 = 0.98$, $p < 10^{-10}$, $n = 34$ in 2017) between UBK and UBD depth. Thus, the UBK determines the vertical location of

the UBD largely. The cold KSSW is denser than water above and the resulting thermocline/pycnocline provides barrier for vertical mixing and below DO depletion can develop. There was an inclination of both clines in both years (Fig. 7). The inclination was particularly steep in 2017. Both isolines were located >50 m depth in the southeastern corner of the study area while in the shallow areas in the west and north the same layer was at 4 – 8 m depth (Figs. 7b,d). Such an inclination of isotherms and cold SST near the coast is an indicator of upwelling.

Deeper colder water intruded along the coastal slope shallower to replace the former upper layer water. The inclination of clines was not that pronounced and it was rather north-south directed in 2015 (Figs 7a,c). Lower correlation in 2015 is caused by discrepancy of the two isolines in the northern part in 2015. This indicates some other processes than Kuroshio water intrusion, e.g. vertical mixing, vertical movement of pycnoclines or advection altered subsurface conditions there.

It is noteworthy that the high vertically integrated AOU (Fig. 6g) area overlapped with shallow position of isolines in 2015 (Figs. 4k, 7a, 7c,). It means that high total AOU there (comparing to neighboring areas) was related both to the thick subsurface DO depleted layer and to the higher (comparing to surrounding area) AOU.

    We have captured two completely distinct situations of stratification and DO distributions in summers 2015 and 2017. The differences between the two years are summarized in Table 1. Prevailing of either pattern in the physical

fields has consequences to the biogeochemical fields and ecosystem in general in the area. The main questions addressed next are: what are the main drivers causing the distinct patterns? What is the forcing behind formation of the patterns?



### 3.2. Forcing, time series

The observed discrepancies of general features in the two cruise surveys can be driven by several forcing mechanisms. The riverine water is a source of buoyancy flux, causing current to flow south along the coast. Mean discharge at Datong hydrological station in the two summers (1 June to 31 August) was rather similar, 44 000 -

45 000 m$^3$s$^{-1}$. We consider the time lag of 1 week for water propagation from Datong to the river mouth (Li and Chen, 2019) in further discussion.

Comparing two years discharge data in June, discharge was higher in 2015. Contrary in July discharge was higher in 2017. Daily discharge maxima peaked at 58 000 m$^3$s$^{-1}$ on 1 July in 2015 and at 70 000 m$^3$s$^{-1}$ on 13 July in 2017. According to the typical residual current velocities off the estuary (Peng et al., 2017) and along the coastline in the south (Wu et al., 2013), it should take a few days to weeks for the water from the river mouth to reach offshore as we registered in 2017 or to the southern part in 2015. We assume that 7 days accumulated run-off (or wind forcing)

is enough to alter the field distributions off the estuary. Mean river flow during the week before surveys was 30 000 m$^3$s$^{-1}$ in 2015 and 65 000 m$^3$s$^{-1}$ in 2017, respectively. Variations in discharge influence the across-shore extent of plume and the width of the buoyant current. According to the theory by Lentz and Helfrich (2002) the width W$_p$ of buoyant current would be 77 km in the case of ambient and CDW water density difference of 10 kg m$^{-3}$, coastal slope of 10$^{-4}$ and discharge of 65 000 m$^3$ s$^{-1}$ before our survey in 2017. The width would be 55 km if the inflow

volume of 30 000 m$^3$ s$^{-1}$ in 2015 was considered. Thus, the variability in the river discharge only somewhat explain the discrepancy between the two surveys, particularly more extensive spreading of the CDW eastward in 2017. However, it does not explain the northeast advection of the CDW that far in 2017. Likewise, it remains unclear why southward transport of the CDW was much more limited in 2017 than 2015. Several indicators hint that wind-driven transport could be behind the discrepancy between the 2015 and 2017 observations, including stronger

across shore inclination of the pycnocline, offshore advection of the CDW, and onshore advection of the colder sub-pycnocline water in 2017. Such features were not found during the survey in 2015, but southward advection of the CDW revealed instead. Since our surveys are conducted annually one month apart, the differences between the two surveys might be associated with the annual cycle in wind climate. Our hypothesis is that the dominant factor behind the discrepancy of the two surveys is wind forcing and possibly its annual cycle.

As argued previously, we assume the thermohaline fields in the study area were established by the forcing one week before. Mean wind, sea level anomaly, surface currents and bottom currents during 7 days prior to the surveys are presented in Fig. 8. The mean wind direction was from northeast prior to the 2015 survey (Fig. 8a) and from south prior to the 2017 survey (Fig. 8b). Northeasterly wind should cause downwelling and accelerate alongshore buoyant current towards the south. This current is clearly visible in the mean surface current plot in 2015 (Fig. 8c),

along the coast from 32.5° N to 28° N. Such coastal current did not reveal before the 2017 survey (Fig. 8d). Weak mean flow towards the east or northeast could be noted near the mouth of the Changjiang River in 2017 (Fig. 8d). Likely, the buoyant coastal current was altered and the CDW was transported across shore by southerly wind forcing. Thus, simulated current patterns and wind forcing prior to the two surveys were qualitatively in accordance with observed salinity fields in the surface layer (Figs. 3a-b). Main difference between the two years in the bottom

layer was the stronger northward flow in 2017 (Figs. 8e-f). Penetration of the deep layer water to the shallower areas, towards the river mouth occurred in 2017 (Fig. 8f). This coincided with our observations in terms of the stronger inclination of clines (Figs. 7b,d) and presence of the colder KSSW water mass in the northern part of the study area in 2017 (Fig. 3h). Because southerly wind forced offshore transport of the surface water, deep layer water intruded to the shallower depths for compensation in 2017. This suggested pattern of processes based on

field surveys and modelling data can be confirmed by mean sea level anomaly during the surveys (Figs. 8g,h). There is strong sea level gradient between near-shore and offshore in both years. Higher sea level can be found along the coast from the northern boundary of study area to 30° N in south in 2015 and as a result, geostrophic

current is southward. This is combined effect of the lighter riverine water and convergence of the shoreward transported upper layer water. This is in accordance with our observation of the downwelling in the central section (Figs. 4b,e,h,k) while such a phenomena did not reveal in the southern section (Figs. 4c,f,i,l) in 2015. Downwelling occurred simultaneously with along-shore southward current in the northern part according to the sea level anomaly (Fig. 8g) and modelled current field (Fig. a,c). This could explain the well ventilated and warmer near-bottom waters in north in 2015 (Fig. 4a,g,j). Opposite gradient, i.e. lower sea level anomaly near coastline were observed in 2017. This is a typical sign of divergence, i.e. offshore transport of the surface water and consequent upwelling along the coastline. The zonal pattern of sea level from the river mouth to offshore (high-low-high sea level, Fig. 8h) agrees with our observation of the upwelling in the central section (Figs. 5b,e). As a result of sea level anomalies, geostrophic current towards north at 28.5-31° N and to offshore near the river mouth were observed in 2017.

In order to verify the sensitivity of the buoyant coastal current to wind forcing, simulated current and wind data from June to September in 1993–2018 were used. First, we calculated daily mean meridional (alongshore) upper layer (0-5 m) current velocity component $v_m$ at the zonal section of 31° N (section S1) from 122.1° to 122.6° E (see red line in Figs. 8c-d). Correlation was calculated between the current velocity component $v_m$ and wind (see the location in Fig. 1) from different directions (full circle by 10° steps). The best correlation ($r^2 = 0.76$, $p < 10^{-10}$, n = 3025) was found with the one-day (prior to the modeled current value) mean SE–NW wind velocity component $w_c$ (Fig. 9). Thus, the meridional current along the coast well correlates with wind.

Simulated near-bottom currents (Fig. 8f) suggest that cold and saltier water (comparing to 2015 observation) was advected to the northern part of the study area as compensation flow to the near-surface offshore flow in 2017. Moderate winds before the 2017 survey (Fig. 10a) could not mix the cold and DO depleted deep layer water into the upper layer. However, stronger wind impulse with speeds up to 11-12 m s$^{-1}$ occurred a few days before the start of the survey in 2015 (Fig. 10a). Upper mixed layer depth created by wind stirring was estimated by the formula describing the turbulent Ekman boundary layer $h = 0.1\ u_* /f$ (Csanady, 1981), where $u_* = (|\tau|/\rho_w)^{1/2}$ is friction velocity, $\rho_w$ is water density, $\tau$ is wind stress and $f$ is Coriolis parameter. Vertical mixing reached down to 15-16 m depth according to this empirical method before the 2015 survey. Note that our estimation is based on 6-h average wind speed and stronger winds, which occur on shorter time scales probably causing deeper mixing. Thus, wind stirring could weaken the stratification before the 2015 survey, as evident by the presence of warmer, fresher water with higher DO concentration in the deep layer. However, also advection from north (Fig. 8c) and downwelling (Fig. 8g) likely contributed to the weakening of stratification and ventilation of the northern part in 2015. Water from north is colder and causes convective mixing and weakening of the thermocline. The effect of downwelling can be well seen in the central section (Fig. 4b,e,h,k). Sea bottom is more flat in the northern part and therefore the sub-thermocline distributions in the case of downwelling showed up there as lateral gradients (Figs. 4a,d,g,j). Closer look to satellite derived daily sea surface temperature (Donlon et al., 2012) and sea level anomaly maps indicate that northern section in 2015 could have been impacted by cold mesoscale cyclonic eddy origin from north. Effect of an cyclonic eddy can be probably seen in Figs. 4a,j. There is lower temperature at distance of 50-60 km with uplifted AOU isolines. Upward lifted isopycnals and oxygen isolines are typical features of cyclonic eddies (Czeschel et al., 2018). We do not investigate further the exact contribution of each process, since it is not under the scope of the present work. However, it is clear, that all the processes evoke by northerly

wind and cause weakening of stratification and vanishing of the colder water mass, which is essential for the hypoxia formation according to our observations in all other sections.

Further we analyzed time series of wind speed (Fig. 10a), SE-NW wind stress component ($\tau_c$, Fig. 10b), meridional current velocity component ($v_m$, Fig. 10c) at the section S1, and the Changjiang river run-off (Fig. 10d) in June-September 2015 and 2017. Wind stress $\tau_c$>0 and northward current component prevailed from June to early August in 2017 (Fig. 10b). Larger wind stress events alternated with calmer periods from June to late August in 2015. During the periods of low $\tau_c$>0, current was often directed southward. The $v_m$ <0 occurred more frequently in June-July 2015 compared to the same period in 2017 (Fig. 10c). Latter is reflected in salinity distributions (Fig. 3a-b). We defined the CDW as salinity of < 30 and calculated its spatial monthly occurrence from June to September in 2015-2018 using satellite surface salinity distributions (Fig. 11). Lowest areal extent of the CDW was observed in September in both years, 2015 and 2017. The CDW mainly advected to the northeast and east in all summer months in 2017. Northeastward transport of the CDW could be noted also in 2015, but the CDW occupied areas in the southern part of the study area as well. Thus, the satellite derived salinity confirms our in-situ observations of southward flowing of the CDW. We can conclude that our survey in 2017 well described the prevailing situation of the CDW spreading in summer 2017. In 2015, we captured the situation, where the CDW spread to the south, but during summer northeastward spreading occurred as well. The CDW spreading in 2016 and 2018 was rather northeastward (similar to 2017). It is noteworthy that coverage of the CDW was clearly larger in 2016 than we observed in 2017 while it was more limited in 2018. The CDW spreading presented in Fig. 11 is further analyzed in relation to inter-annual wind and river forcing, and in the context to previous studies in the discussion.

Next we make an attempt to quantify the two CDW spreading cases caused by wind forcing through analyzing meridional currents $v_m$ at section S1 (see Fig. 8c-d) and wind data from the period June–September in 1993-2018. We grouped the simulated $v_m$ to the wind velocity $w_c$ classes with step of 0.25 ms$^{-1}$ and took average of the each group. By doing this, we found relation between mean $w_c$ and $v_m$. We found that if the daily mean wind velocity component $w_c$=0 m s$^{-1}$, current velocity component $v_m$ would be on average -13 cm s$^{-1}$ (i.e., southward) at S1. We can assume that this is the mean geostrophic component of the current velocity. Daily mean wind velocity component $w_c$ of 5.6 m s$^{-1}$ is required to reverse the current and have the same magnitude at S1 ($v_m$ = +13 cm s$^{-1}$). This corresponds roughly to the wind stress $\tau_c$ of 0.04 N m$^{-2}$. Wind stress $\tau_c$ of 0.02 N m$^{-2}$ corresponds to the $v_m$ of 0 cm s$^{-1}$, i.e., coastal current is altered. There are two main wind-driven processes that contribute to the alteration of the coastal current. First is the direct effect of wind stress and resulting Ekman surface current. Secondly, southerly winds cause divergence and upwelling at the coast and alter sea level distributions. When upwelling is evoked, alongshore geostrophic current, resulting from the cross-shore gradient between colder-denser and warmer-lighter offshore water, is rather to north. The mean sea level anomalies during our surveys in 2015 and 2017 illustrate well this phenomenon (Fig. 8g-h). The joint effect of the two processes explain relatively low wind stress required for the alteration of the circulation regime. We analyzed the simulated current data in June–September from 1993 to 2018 and averaged the cases, when daily mean wind stress component $\tau_c$ was < 0 and ≥ 0.02 N m$^{-2}$ (Fig. 12). The same circulation features, but more pronounced, stand out as before the surveys in 2015 and 2017 (Fig. 8). In the case of $\tau_c$ < 0 N m$^{-2}$ southerly flow prevailed in the surface layer along the coast (Fig. 12b) and the mean currents in the near-bottom layer were rather weak (Fig. 12d) as before our survey in 2015 (Figs. 8c,e). In the case of $\tau_c$ ≥ 0.02 N m$^{-2}$ surface flow was northward along the coast, to northeast near river

mouth (Fig. 12a) while shifted north or towards the river mouth in the deep layer (Fig. 12c) as before our survey in 2017 (Fig. 8d,f). Thus, latter supports our hypothesis of the wind-induced simulated current patterns in 2015/2017 (Fig. 8a-f).


**Discussion**

The main reason for the oxygen consumption and concurrent hypoxia is decomposition of organic matter by
bacteria in the sub-surface layer. There is a strong link between primary production in the upper layer and oxygen consumption in the near bottom layer (Chen et al., 2017, 2004; Wang et al., 2017, 2016; Wei et al., 2017a; Zhou et al., 2019). The processes accompanied with enhanced primary production, spreading of the riverine nutrient rich CDW and KSSW upwelling, are dependent on wind and river forcing. The effects of wind and river discharge on the spatiotemporal variability of stratification and oxygen depletion will be discussed next.
Two distinct stratification, Chl $a$ and DO distribution patterns were observed in the area off the Changjiang Estuary in the summers of 2015 and 2017. There was a stronger DO depletion in the southern part of the study area in August-September 2015, while intense hypoxia was observed in the northern part of the study area in July 2017. Likewise, remarkable DO depletion was observed in the southern part in the upwelling region in 2017. Similar hypoxia patterns in August (our July 2017 observations) and October (our August-September 2015 observations)
were observed in 2006 (Zhu et al., 2011). The main reasons for the different spatial hypoxia patterns are variations in the wind forcing and river discharge. Summer monsoon and higher discharge both favor across-shore transport of CDW and development of hypoxia in the northern part of the study area. Both, maximum frequency of southerly wind and river discharge in the annual cycle occur in July-August (Figs. 13a,c). Thus, our observations conducted in late August - early September 2015 and late July 2017 illustrate the annual cycle of forcing and latter reflect in
oxygen and stratification patterns. On the other hand summer monsoon and river discharge were close to average in the whole summer of 2017 while the summer monsoon was clearly weaker in 2015 (Fig. 13a-b). Thus, our observations reflect also the differences in forcing and concurrently the DO distributions between two summers, i.e. inter-annual difference. In accordance with the annual cycle of wind (Fig. 13a), highest frequency of upwelling events occurred in July-August as well (Xu et al., 2017). Wang et al. (2012) explained the annual cycle of the
location of near-bottom DO minimum by seasonal change of stratification due to river discharge and warming/cooling of the upper layer. We suggest that the southern location of the DO minimum in June and September, and the northern location, as observed in July-August 2006 (Fig. 3 in Wang et al. (2012) paper), are related also to the seasonality of wind forcing, respective surface current and compensating KSSW intrusion in the deep layer.
In figure 11, we presented monthly occurrence of CDW (as salinity < 30) from June to September 2015-2018. It is clear from fig. 11 that CDW transport offshore (to northeast) has occurred in all years, including 2015. However, one can see how year 2015 differs with low value in the inter-annual time series of wind stress of $\tau_c \geq 0.02$ N m$^{-2}$ (Fig. 13a-b) and with considerable southward advection of the CDW (Fig. 11). Monthly mappings of bottom oxygen in 2015 (Li et al., 2018b) does not show significant oxygen depletion in north in any month while

deteriorated hypoxia (comparing to our observations) in south occurred in October. This well demonstrates that wind regime plays an important role in the inter-annual variations of the hypoxic area location (Fig. 6e-f).Year 2016 stands out in terms of wind forcing close to inter-annual mean (Fig. 13b), but with very high river discharge (Fig. 13d). One can see that offshore advection prevailed in 2016 (Fig. 11). This indicates that if wind forcing is close to long-term average hypoxia more likely will occur rather in the east of the river mouth (northern part of

study area) and hypoxia occurrence in the south is more limited. This can be confirmed by our observations in 2017. River discharge and $\tau_c \geq 0.02$ N m$^{-2}$ occurrence were both similar to long term mean in 2017 (Fig. 13) and we observed hypoxia and CDW water in the northern part. It means, hypoxia in the southern part (as we observed in 2015) occurs if the summer monsoon is considerably weaker than long-term mean. Indeed, hypoxia in the south has been quite rare in 1998 – 2013 (Chen et al., 2017). As a consequence of large river runoff, clearly largest CDW

spreading area occurred in year 2016 out of the four years (Fig. 11). Occurrence of wind stress $\tau_c \geq 0.02$ N m$^{-2}$ was higher than average (Fig. 13b) and river discharge was lower than average in 2018 (Fig. 13d) and this reflected well in the CDW distribution (Fig. 11). One can see that CDW covered smallest area out of four years and prevailing transport direction was towards the north or northeast (Fig. 11). From our stratification and DO distribution observations (Figs. 3-7) and their dependence on wind forcing (Figs. 8-10, 12), and CDW spreading

direction and extent (Fig. 11), DO depleted area could have occurred in the northern part both in 2016 and 2018, with larger extent than in the former year. Hypoxia in the northern part in 2016 has been reported (Zhang et al., 2019). Lower than average river discharge and close to average occurrence of $\tau_c \geq 0.02$ N m$^{-2}$ (as in 2016) occurred in 2006 (Figs. 13b,d), when hypoxia registered (Zhu et al., 2011).

  We estimated that wind stress of $\tau_c \geq 0.02$ N m$^{-2}$ is necessary to alter the geostrophic coastal current and create

favorable conditions for hypoxia in the northern part. Inter-annual variations of occurrence of wind stress $\tau_c \geq 0.02$ N m$^{-2}$ in July-August are quite large (Fig. 13a). When the occurrence of $\tau_c \geq 0.02$ N m$^{-2}$ was $\geq 50\%$ in the summers of 1995, 2012, 2013, and 2018; stratification and DO patterns could have occurred as coastal upwelling-related hypoxia in the south and large hypoxic area in the north as we observed in 2017 (Fig. 6b,f,h). Large hypoxic area in the north have been documented for instance in August 2012 (Li, 2015) and August 2013 (Ye et al., 2016; Zhu

et al., 2017). Contrary, in summers when the occurrence of $\tau_c \geq 0.02$ N m$^{-2}$ was less than inter-annual mean (1992, 1993, 1996, 1998, 1999, 2002, 2007-2009, 2014), considerable oxygen depletion was also expected to have occurred in the southern part as we observed in 2015. Such situation has been captured for instance in 1998 (Wang and Wang, 2007) and in 1999 (Li et al., 2002). However, on the top of the inter-annual variability and annual cycle are synoptic scale changes of wind-driven currents and river forcing, which likely influence the distributions on

shorter time scales (Fig. 10). Thus, when planning hypoxia related measurement campaigns in future, it is worthwhile to take into account wind-driven transport, river discharge, remotely sensed salinity and altimetry to forecast spreading of the CDW, upwelling occurrence and deep water intrusion and according to latter factors estimate potential hypoxic area prior to field works. This could allow more efficient use of ship time and more detailed sampling of the hypoxic area.


  The faith of the river plume can be separated to the regions and processes: circulating bulge near the mouth and downstream current along the coast (Fong and Geyer, 2002; Horner-Devine, 2009). The question is how much of riverine water remains in the river plume bulge and how much is advected to the neighboring areas. It has been estimated  that about 80-90% of the discharge accounts to freshwater transport of coastal current (Li and Rong,

2012; Wu et al., 2013). This means that most of the river discharge does not remain in the river plume bulge, but impacts the surrounding areas off the estuary. Applying linear regression analysis between wind and modeled current velocity we estimated the southward geostrophic component of the current velocity to be on average 13 cm s$^{-1}$ at section S1. The southward coastal current was measured in winter by Wu et al. (2013). They estimated the maximum detided current speed up to 50 cm s$^{-1}$, but it included also wind-driven component, which in winter

supports flow to the south (Wu et al., 2013).

Offshore, east- or northeastward advected CDW caused by southerly wind, as we observed in 2017, might form detached eddies due to interaction of the Ekman flow and density driven frontal currents (Xuan et al., 2012). Those eddies bring CDW further offshore and alter physical, chemical characteristics (including oxygen conditions) and primary production in the water column (Wei et al., 2017). On the other hand we noted ventilating impact of

colder cyclonic eddy in north in 2015.

Offshore transported CDW occurred simultaneously with coastal upwelling in 2017, as both processes require southerly wind. Shoreward, upslope penetration of the sub-thermocline KSSW and hypoxia in the upwelling – CDW coupling zone (Wei et al., 2017a) were observed. Upwelling, induced by southerly winds and its relaxation supported by northerly winds was captured by cross-sectional in-situ measurements by Yang et al. (2019).

Idealized numerical experiment by Liu and Gan (2014) showed that southeasterly wind forcing caused the development of upwelling and shoreward intrusion of colder water in the study area. Wei et al., (2017) showed how the coupling of the CDW plume front and KSSW upwelling caused DO minimum at the sloping bottom. Likewise N/P ratio and primary production in the CDW is considerably modified by upwelling (Tseng et al., 2014). Phosphate transport by upwelling reduces phosphorus deficiency in the CDW water and therefore promote

phytoplankton growth and nitrate uptake (Chen et al., 2004; Zhou et al., 2019).

Time series of wind displayed large variations in wind forcing on shorter time scales (days to weeks) (Fig. 10) which may alter the stratification pattern and DO distributions considerably. Numerical simulation by Zhang et al. (2018) showed that wind-induced redistribution of the Changjiang River plume changes near-bottom DO conditions rapidly. Also, it has been shown that vertical mixing caused considerable variations in DO

concentrations in the near-bottom layer (Ni et al., 2016). In July 2015, 1.5 month before our survey, hypoxia was terminated by typhoon, but two days later hypoxic conditions were re-established (Guo et al., 2019). Our field measurements showed that UBD is strongly linked to the UBK subsurface water (Fig. 7). Besides enhanced (or impeded) vertical diapycnal mixing, DO conditions can be altered by vertical movement of this water mass. Ni et al. (2016) published a valuable dataset of time series DO data in the near-bottom layer. They linked the increase

of DO concentrations in the near-bottom layer with the vertical mixing and DO decline in the near-bottom layer with the primary production and consequent decomposition of detritus. One can see several cases from their time series data (Fig. 2 in Ni et al. (2016)), when near-bottom temperature drops. Those events must be related to the advection of colder water and uplift of the KSSW. At the same time, DO declined during those events. Penetration of the cold, low DO water upwards along the coastal slope appear as temperature and DO decline in the point

measurement time series. On the other hand there were some events, where near-bottom layer temperature rises, but sea surface temperature does not change much (e.g. in the beginning of their time series, Fig. 2 in Ni et al. (2016). In these cases, DO concentrations increased in the near-bottom layer. Such events can be associated rather to the downward movement of the thermocline or advection of warmer water as we observed in the central section in 2015 (Figs. 4b,e,h,k), than vertical mixing. Thus, vertical location and movement of the thermocline has

important role in the near-bottom DO distributions at the coastal slope. Importance of KSSW thickness on the oxygen depletion estimations reveal well also if near bottom oxygen maps are compared with the total AOU maps (Figs. 6g-h). Bottom hypoxia in north in 2017 was much more intense comparing to hypoxia in south in 2015. However, the total AOU was similar in hypoxic zones in both years due to thicker oxygen depleted layer, i.e. thicker KSSW in south in 2015.

Pycnocline dynamics, including downwelling studies, as suggested by Hu and Wang, (2016), are important in future investigations. Those studies are difficult to arrange with conventional research vessel surveys only. Autonomous measurement platforms, such as profiling moorings (Lips et al., 2016; Sun et al., 2016), moored sensor chains (Bailey et al., 2019; Venkatesan et al., 2016), which allow capturing the variability in necessary spatio-temporal resolutions can be used. Underwater gliders (Liblik et al., 2016; Rudnick, 2016) might be

complicated to use due to strong tidal velocities and heavy ship traffic, but are worthy to consider as well.

Our observations indicated that two conditions in the water column must be present for the development of hypoxia. First, high AOU occurred only below thermocline, in the cold KSSW. Secondly, primary production in the fresher CDW or in upwelling is needed to cause high DO consumption that leads to hypoxia. Interestingly, these two conditions for hypoxia were valid in very different situations. Shallow (4-6 m) and sharp thermocline

coincided with the halocline (related to the CDW) in the northern part of the study area in 2017. Contrary, thermocline and halocline were clearly separated in the southern part of the study area in 2015. Thus, vertical coincidence of halocline and thermocline is not necessary for the hypoxia formation. The thermocline acts as physical barrier, which impedes vertical mixing and DO exchange with upper layers. Primary production in the CDW or in the upwelled water and related DO consumption in the near-bottom layers (Wang et al., 2017, 2016)

cause DO decline. Zhu et al (2016) found that the area of DO concentration $<3.0$ mg l$^{-1}$ fits relatively well with the region of the pycnocline strength $> 2.0$ kg m$^{-3}$. This relationship between pycnocline strength and low DO only partly holds truth according to our mappings (Figs. 6a-b, e-f). For instance, there was strong stratification (Fig. 6a) near the river mouth (around 122.5° E and 31° N) in 2015 due to presence of the CDW, but hypoxia was not observed (Fig. 6e) since the colder KSSW was not present. Most of the study area, except very shallow areas had

strong enough stratification (pycnocline strength $> 2.0$ kg m$^{-3}$) in 2017 (Fig. 6b). However, hypoxia in the southern part in 2017 was only observed near the coast and in connection to the coastal upwelling (Fig. 6f). Rest of the study area in south was free of hypoxia. In the latter area KSSW was present, but there was no CDW in the surface layer. This means strong stratification as such does not lead necessarily to hypoxia. Two features must be present for hypoxia formation: 1) KSSW, 2) CDW and/or subsurface water upwelling. We can conclude that colder KSSW

determines where (including in what depths) hypoxia could develop. Thus, latter provides necessary precondition for hypoxia. The CDW spreading and/or surbsurface water upwelling (and related biogeochemical, biological processes) determine the magnitude, exact location and timing of oxygen depletion. Both features are strongly impacted by wind.

Besides the barrier effect by creation of the thermocline, intrusion of KSSW has other implications on oxygen

dynamics. First, the subsurface water is oxygen depleted already before local impact of oxygen consumption. The furthest stations in the southeast (Fig. 1) had AOU of 2-2.5 mg l$^{-1}$ in the deep layer in 2017, i.e. in the same order that has been estimated in the KSSW before (Qian et al., 2017). The total AOU in the water column there was 50-60 g m$^{-2}$ (Fig. 6h). This water is still rather well ventilated comparing to the deep layer waters that had been impacted by upwelling or CDW induced production in the surface layer (Fig. 6g-h). Despite its initial oxygen

depletion, Kuroshio intrusion is important source of oxygen import to the study area (Zuo et al., 2019). Without this lateral oxygen advection, hypoxia could form much faster in larger area (Zuo et al., 2019). Kuroshio intrusion is nutrient rich (Zhang et al., 2007b; Zhou et al., 2019) and its upwelling or vertical mixing could intensify sequence of primary production in the surface, consequently organic matter sinking producing oxygen consumption in the near-bottom layers.

We have already outlined the main difference in the wind forcing behind formation of the hypoxia in the southern and northern parts of our study area. The one in the north develops under the conditions of summer monsoon. Intense hypoxia can start from very shallow depths (at 4-8 m) in the northern area (Fig. 5g) and it can develop very fast under favorable conditions (Guo et al., 2019). On the other hand, bottom layer there can be easily ventilated by wind stirring or downwelling as we observed at N15 section in 2015 (Fig. 4g). Thus, hypoxia in the northern

part can be very pronounced, but disappears fast if forcing changes (Ni et al., 2016).
        Hypoxia in the southern part of the study area was not so pronounced but quite stable as noted already by Zhu et al. (2011). Continuous decline of the DO from March to October in the southern part was well demonstrated by monthly measurements (Li et al., 2018b). We suggest three main reasons for this. First, favorable wind conditions for the southward transport are not that frequent in summer. Secondly, the mixture of CDW with ambient ocean

water promotes organic matter settling and nutrients consumption on the way to south, so less detritus sinks to the bottom layer and DO consumption is lower in the southern part comparing to the northern counterpart. Third, the KSSW and related thermocline are located deeper in the south and therefore wind mixing does not destroy the thermocline. In short, the first two reasons account for a lighter hypoxia and the third for a long lasting hypoxia in the southern part.


**Conclusions**

Two main conditions in the water column must be present for the occurrence of hypoxia in the near-bottom layer off the Changjiang Estuary: enhanced primary production in the upper layer and KSSW intrusion in the near bottom layer. Pycnocline created by KSSW intrusion is precondition and determines where hypoxia could develop. Primary production in the CDW and/or in the upwelled water, and consequent oxygen consumption by sinking organic matter below pycnocline determine the magnitude, exact location and timing of oxygen depletion.

Advection of the CDW and KSSW, likewise occurrence of upwelling, are strongly related to the wind forcing. Summer monsoon (wind from south) alters CCC by creating Ekman surface flow and by changing the geostrophic current flowing to north or northeast. As a result, the CDW spreads offshore, and KSSW intrudes northwards and upwards on coastal slope, consequently producing upwelling. Joint effect of these processes can lead to intense and shallow (4-8 m) oxygen depletion in north and hypoxia at coastal slope in south.

Northerly wind intensifies CCC and CDW spreading to south, and causes downwelling. As a consequence, northern part is well ventilated and hypoxia rather occurs further offshore in the southern part.
        Wind forcing and river runoff are important contributors of inter-annual variations and annual cycle, determining the size and location of low DO areas. The DO minimum is located more likely in the northern part in July-August and in the southern part during rest of the stratified period.

There is a strong connection between the upper boundaries of KSSW intrusion and oxygen depletion. The sensibility of the boundaries to wind forcing shapes oxygen conditions considerably in the area. Autonomous measurement campaigns by mooring arrays and underwater gliders could considerably improve the knowledge about related processes. Concepts suggested in the present work can be utilized, when planning in-situ experiments. Wind, river discharge, remotely sensed salinity and altimetry data can be used to forecast hydrographic situation and potential hypoxic areas prior to field works.

*Code availability.* Scripts to analyze the results are available upon request. Please contact TL.

*Author contributions.* TL lead the analyzes of the data and writing of the manuscript with contributions of DF and YW. DF was responsible to arrange oceanographic cruises. DS measured chlorophyll *a* in 2015 cruise.

*Competing interests.* We declare that no competing interests are present.

*Acknowledgments.* This research was funded by the National Natural Science Foundation of China (41776052), Qingdao National Laboratory for Marine Science and Technology (MGQNLM-TD201802), and the Research Fund of State Key Laboratory of Marine Geology at Tongji University (MG20190104).
River data was downloaded from Bulletin of River Sediment in China provided by Ministry of Water Resources of the People's Republic of China (MWR). Website: http://www.mwr.gov.cn/sj/tjgb/zghlnsgb/ (visited 7th September 2019). We thank Yue Zhang for gathering CTD data in 2015, Junbiao Tu for providing mooring data, Huiping Xu for agreeing to use their chlorophyll *a* data. We would like to thank our colleagues who helped us in performing measurements. We thank Jaan Laanemets for his comments on the manuscript. We thank reviewer Fabian Große and anonymous reviewer for the valuable comments and suggestions that helped to improve the manuscript.

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

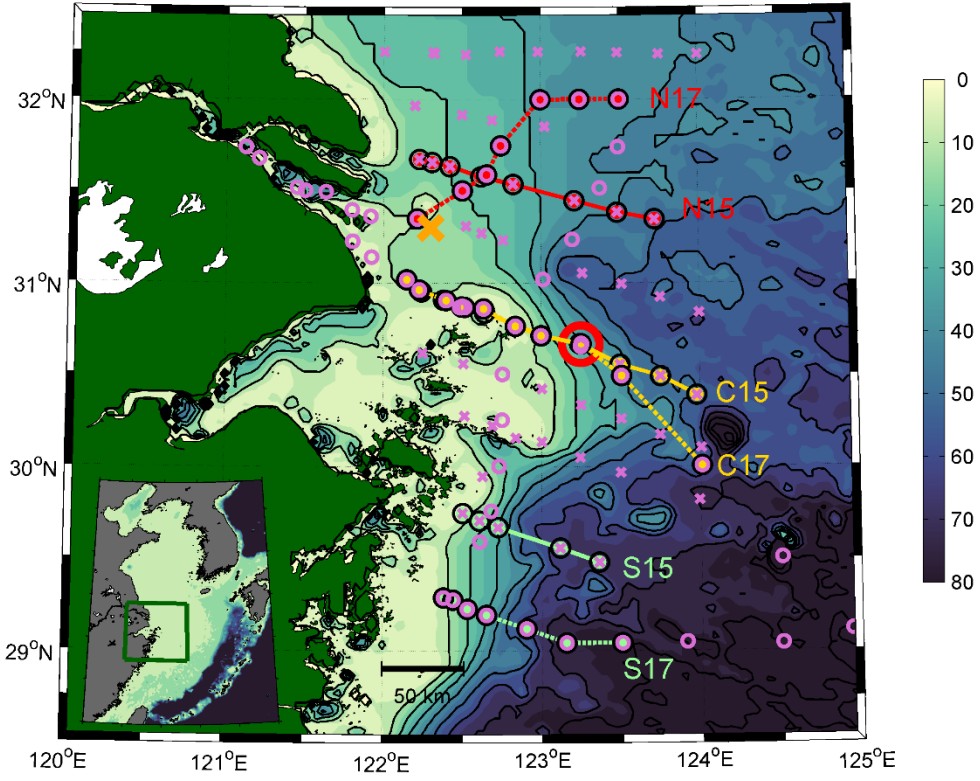

**Figure 1. Map of the study area off the Changjiang Estuary. Crosses represent CTD cast stations conducted on 27 August – 5 September in 2015 and circles show CTD cast stations on 24 - 29 July in 2017. Lines represent northerly sections (N15 in 2015 and N17 in 2017), central sections (C15 in 2015 and C17 in 2017), and southern sections (S15 in 2015 and S17 in 2017). Larger circle represents the mooring M1 location. Larger cross shows the location, where wind data were gathered. Color scale shows depth (m) of the study area. The inlay shows the study area in the East China Sea.**

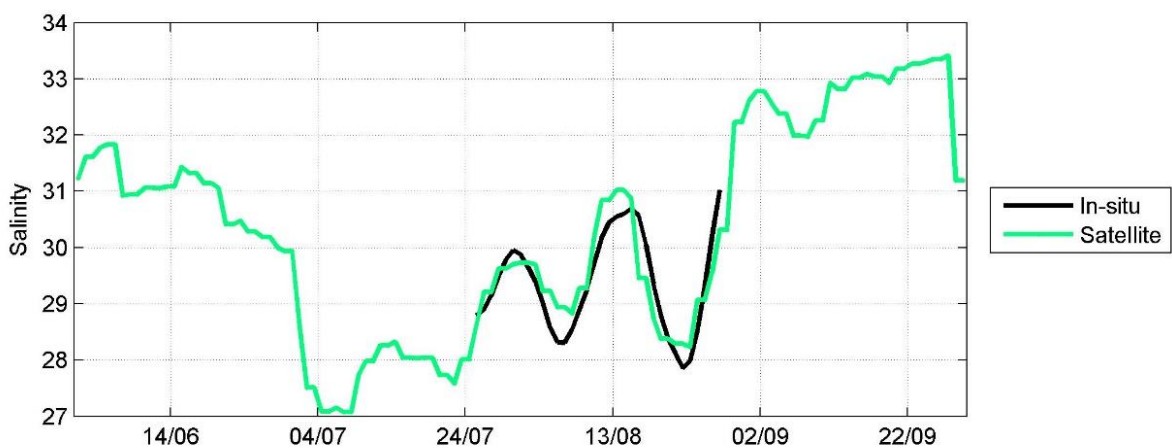

**Figure 2. Time series of in-situ and remotely sensed 8-day running mean salinity at the M1 location in 2017.**

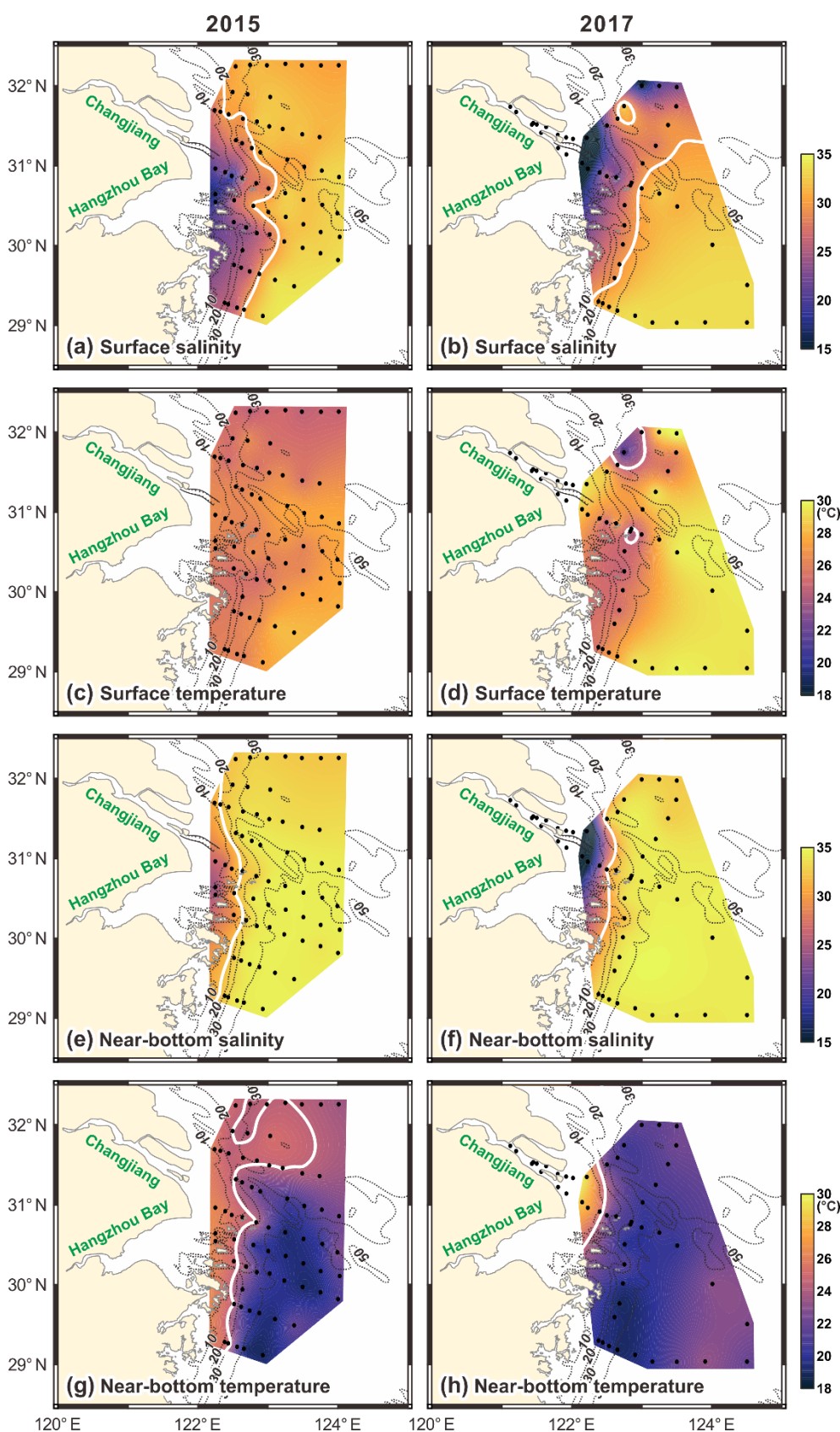

**Figure 3. Maps of surface salinity (a, b), surface temperature (c, d), near-bottom salinity (e, f) and near-bottom temperature (g, h) from the surveys in 2015 (left panel) and 2017 (right panel). 24.5 °C and 30 isolines are shown as white lines.**


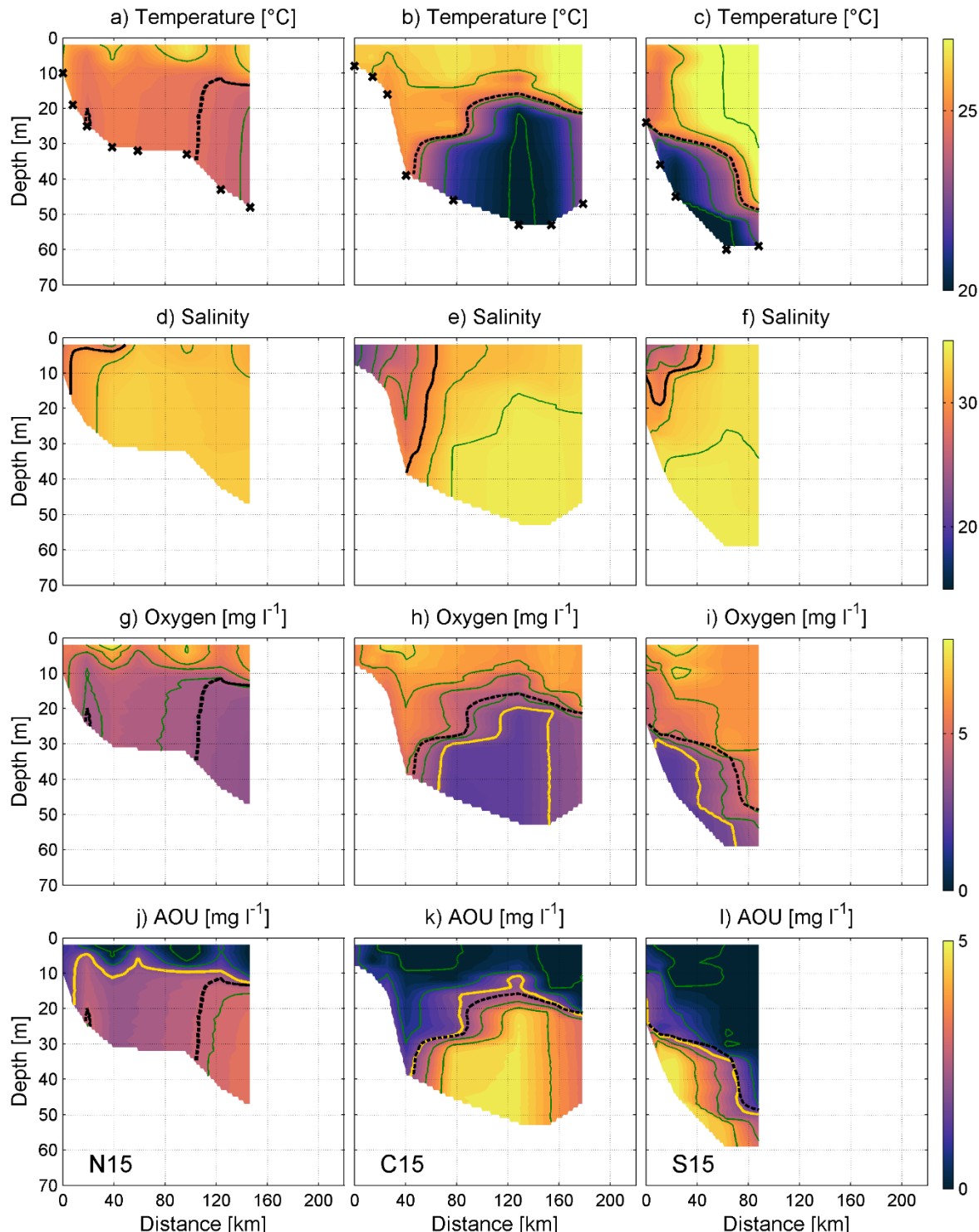

**Fig. 4. Vertical distributions of temperature, salinity, density, DO concentration, and Apparent Oxygen Utilization (AOU) along sections N15, C15, and S15 in 2015 (Fig. 1). Temperature isoline 24.5 °C is shown with thicker dashed line in the temperature, oxygen and AOU plots. AOU isoline 2 mg l⁻¹ is shown with thicker solid line in AOU plots. Hypoxia border (3.0 mg l⁻¹) is shown with thicker solid line in oxygen plots. Thin lines represent isolines of temperature with 2 °C step, salinity with 2 step, oxygen and AOU with 2 mgl⁻¹ step both. Locations of CTD stations are shown as crosses in the top panels. Values on the *x*-axis indicate the distance from the westernmost point of a section (Fig. 1).**


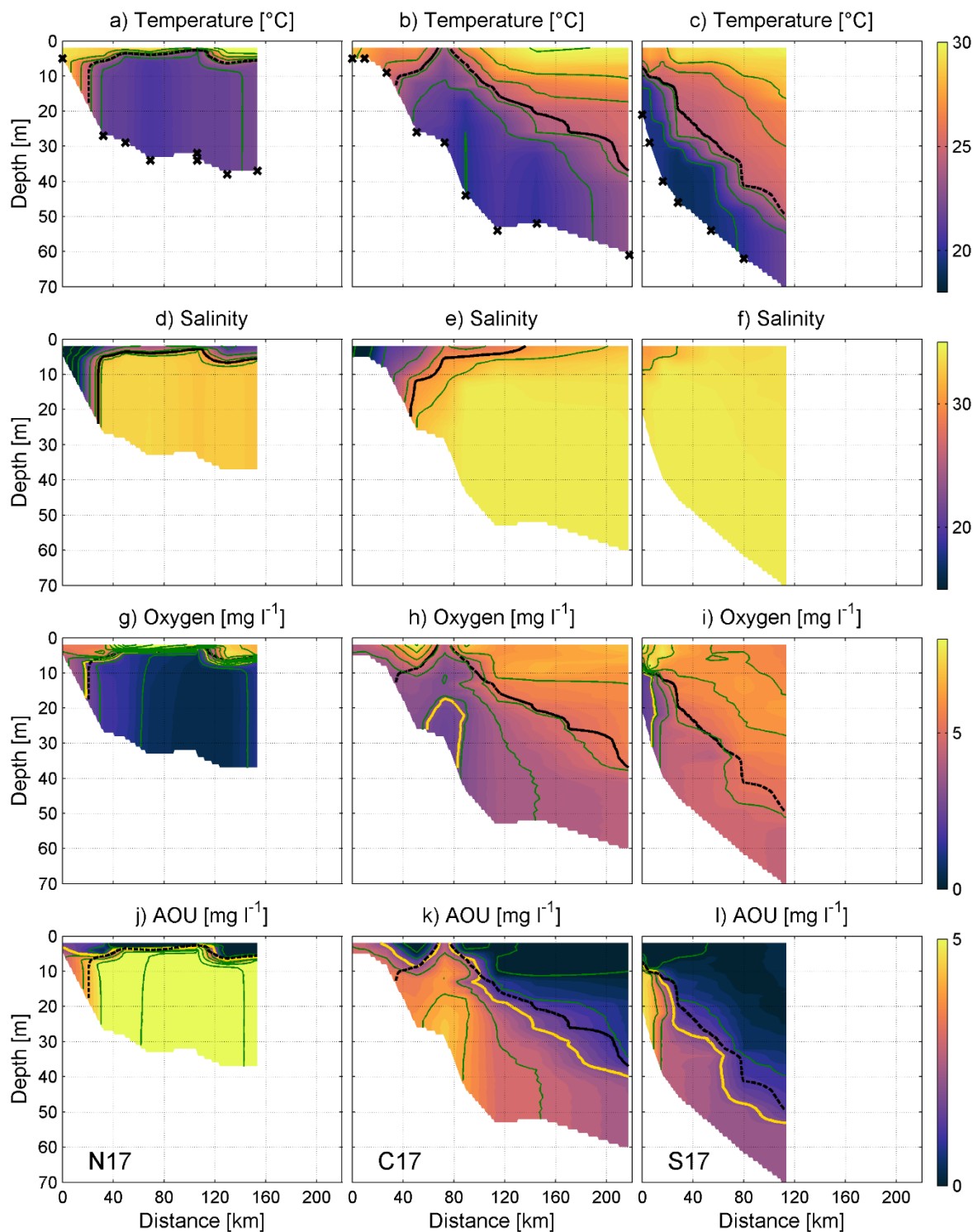


**Fig. 5. Vertical distributions of temperature, salinity, density, DO concentration and AOU along sections N17, C17, S17 in 2017 (Fig. 1). Temperature isoline 24.5 °C is shown with dashed line in the temperature, oxygen and AOU plots. AOU isoline 2 mg l⁻¹ is shown with solid line in AOU plots. Hypoxia border (3.0 mg l⁻¹) is shown with solid line in oxygen plots. Locations of CTD stations are shown as crosses in the top panels. Thin lines represent isolines of temperature**
**with 2 °C step, salinity with 4 step, oxygen and AOU with 2 mgl⁻¹ step both. Values on the *x*-axis indicate the distance from the westernmost point of a section (Fig. 1).**

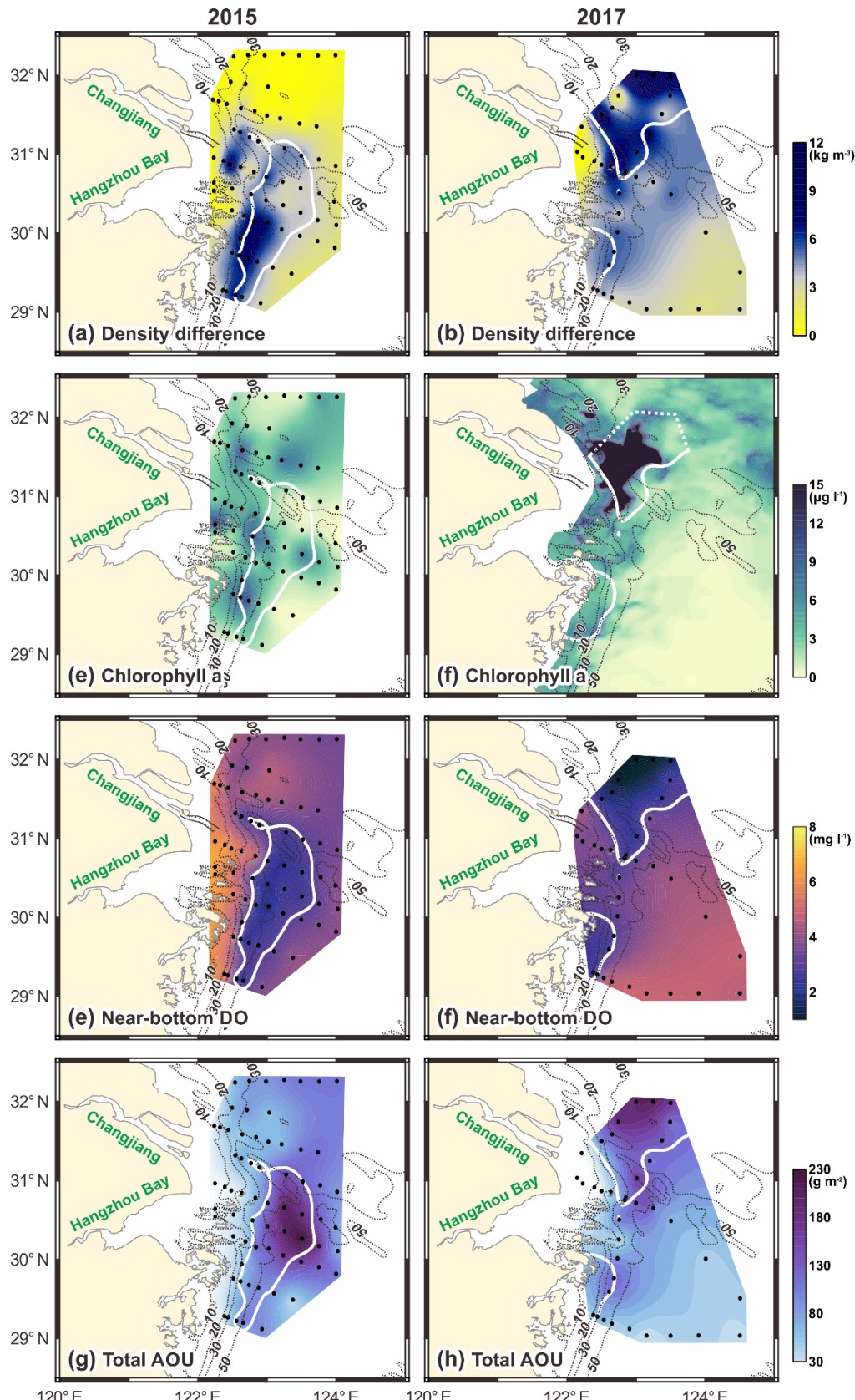

**Fig 6. Maps of density difference between the near-bottom layer and surface layer (a,b), in-situ Chl *a* (c), satellite derived Chl *a* (d), near-bottom DO (e,f) and total AOU in the water column (g,h). Maps of 2015 are shown on left panels and 2017 on right. Remotely sensed Chl *a* map (f) is a mean field of two daily images of 21-22 July. AOU was calculated according to Eq. (3).**

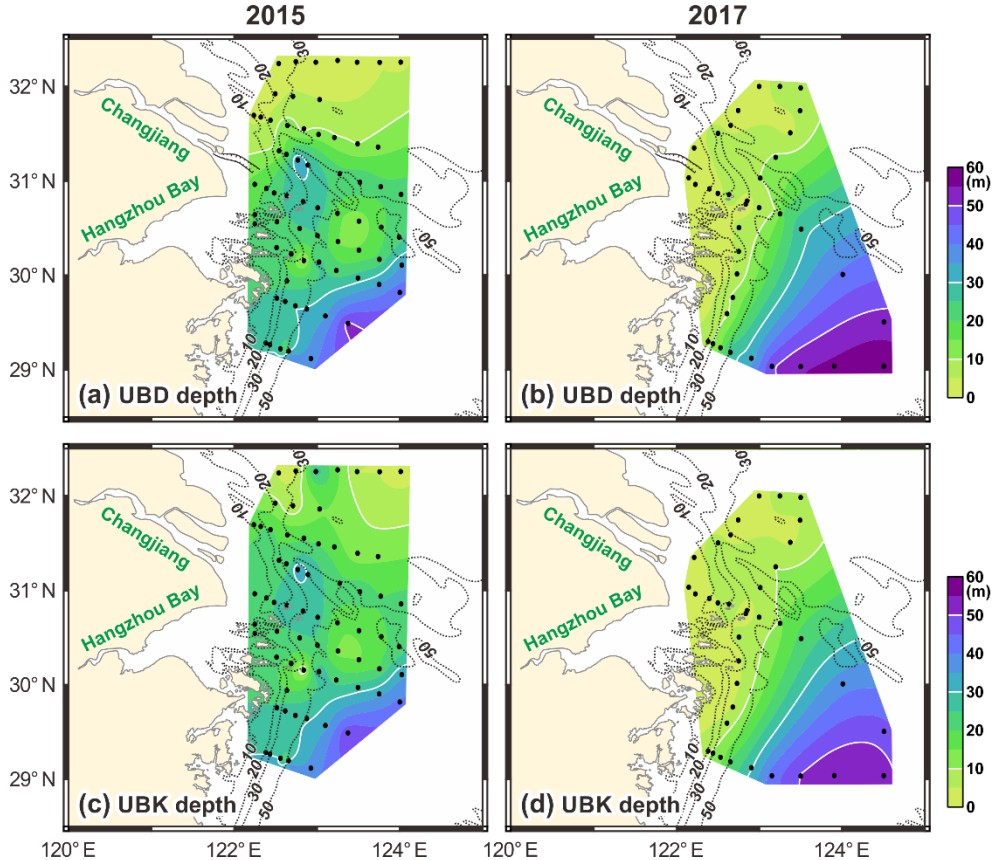

**Fig. 7. Maps of AOU isoline 2 mg l⁻¹ (UBD) depth (upper panels) and temperature isoline 24.5 °C (UBK) depth (lower panels) in 2015 (left panels) and in 2017 (right panels).**

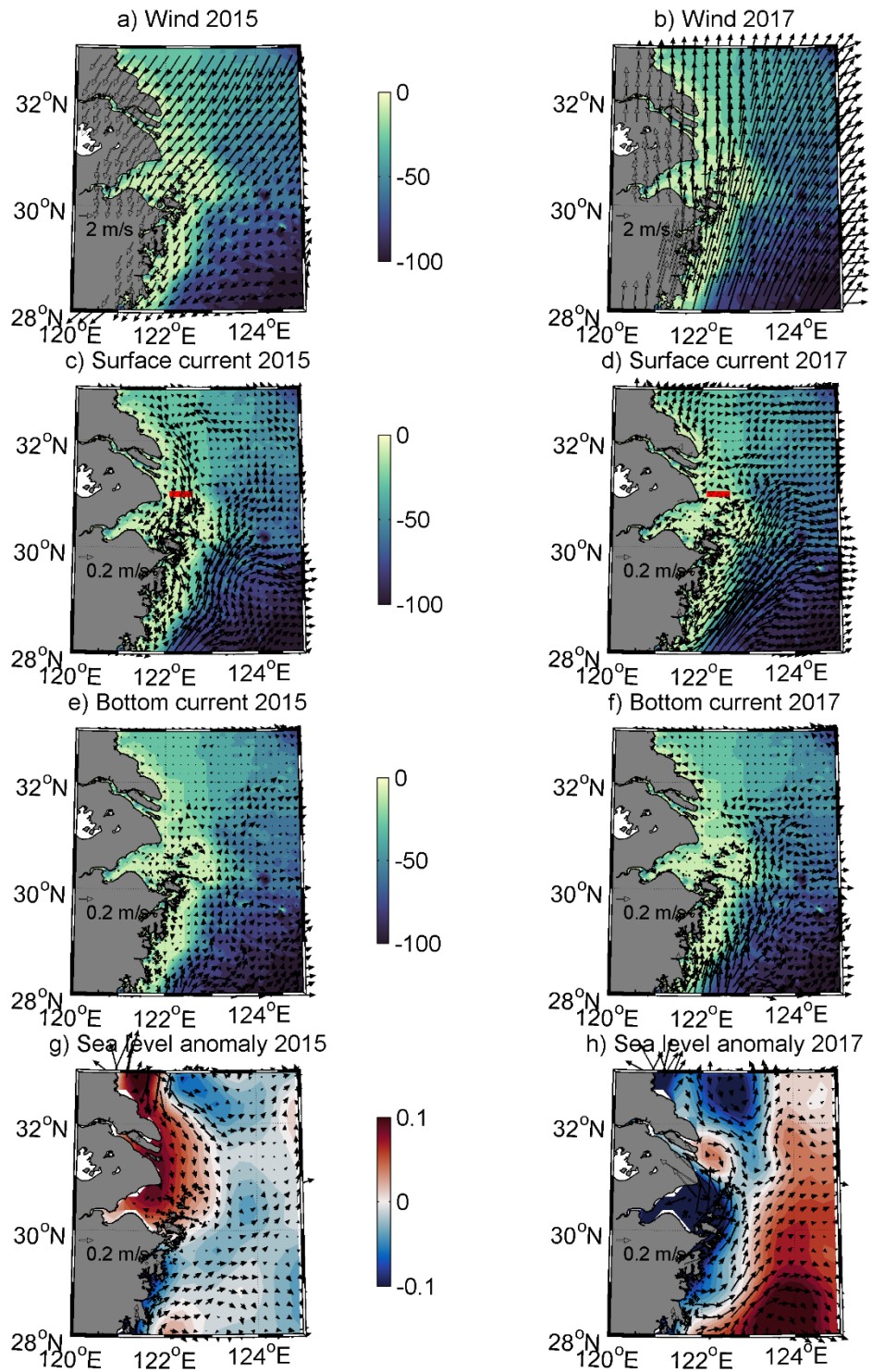


**Figure 8. Maps of mean wind (panels a-b, ms⁻¹), simulated surface current (panels c-d, cms⁻¹) and bottom current (e-f, cms⁻¹) s currents in the sea bottom depths down to 100 m during 7 days period prior to the CTD surveys. Bottom bathymetry is shown as a background in panels a-f. Red lines in panels c and d show the section S1, where current time series (presented in Fig. 10c) were calculated. Every fourth current vector is shown only in the panels c-f. Mean sea level anomaly (m) as contours and geostrophic velocities (cms⁻¹) as vectors derived from satellite altimetry during the surveys are shown in the panels g-h.**


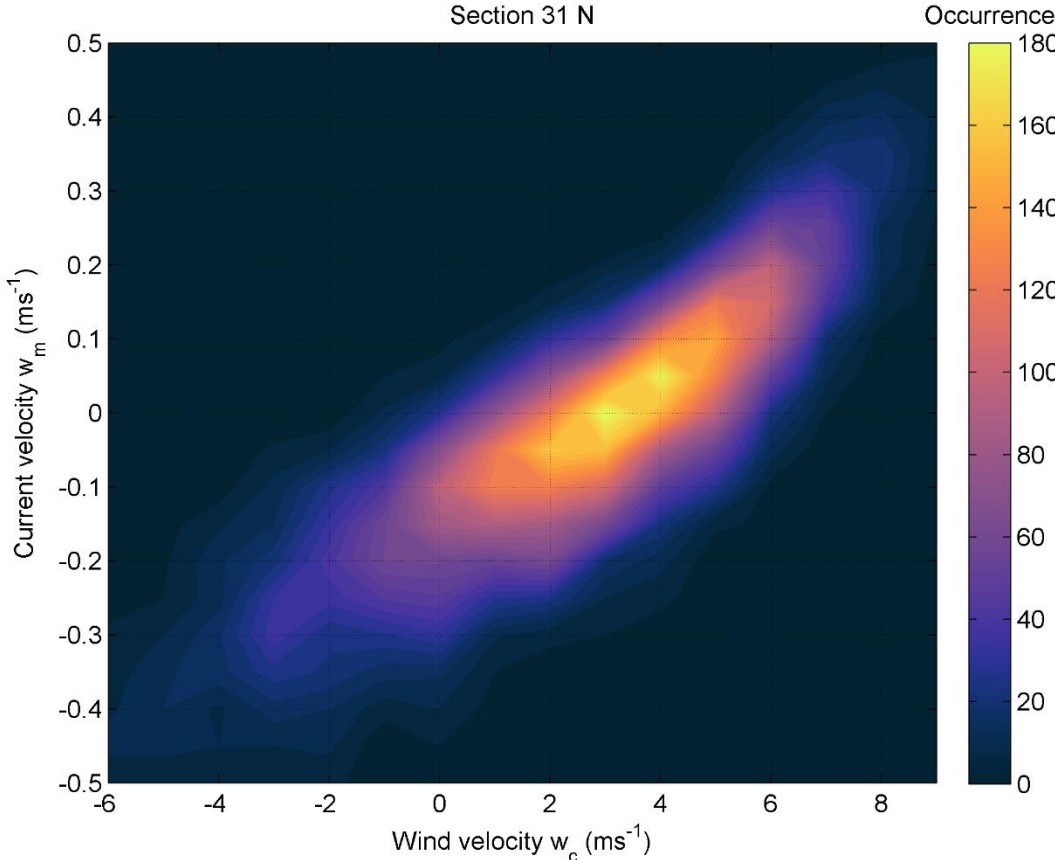

**Figure 9. Relation of daily mean SE-NW wind velocity component *w*c (positive towards NW) and meridional current velocity component *v*m (positive northward) in the section at 31° (section S1 in Fig. 8) in 1993-2018. Color scale shows the co-occurrence (number of cases) of respective wind velocity and current velocity component combinations.**


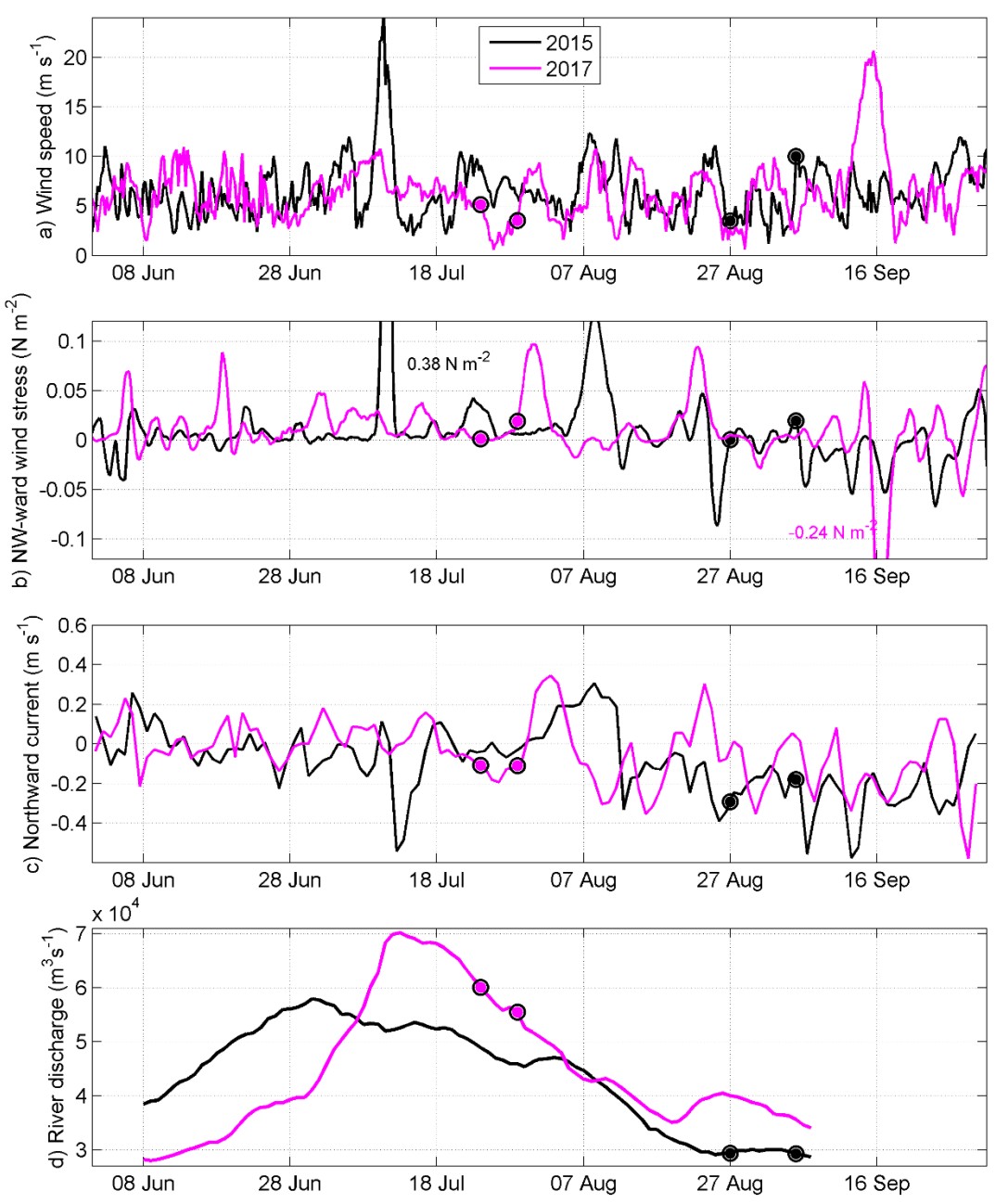

**Figure 10. Time series of wind, current, and river discharge. Mean wind speed (6-hours) (a), 1-day running mean SE-NW wind stress $\tau_c$ (positive towards NW; N m$^{-2}$) (b), daily mean meridional current velocity component $v_m$ (positive northward) at section S1 (c), and daily Changjiang river discharge at Datong station shifted by 7 days to represent flow in the river mouth (d). Blue and red dots represent start and end of the CTD surveys, respectively in 2015 and 2017.**


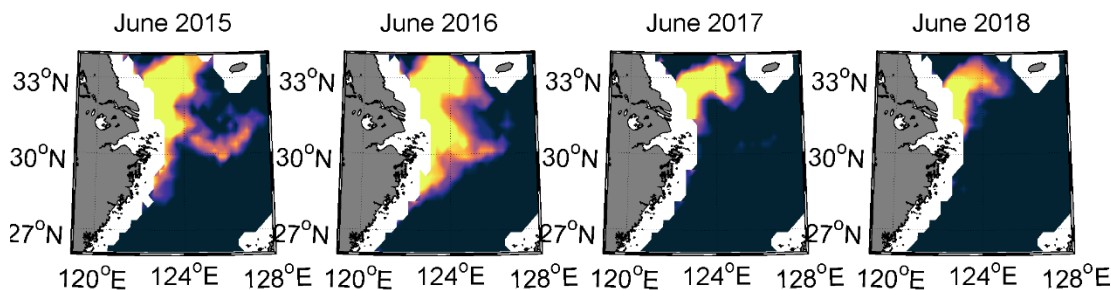

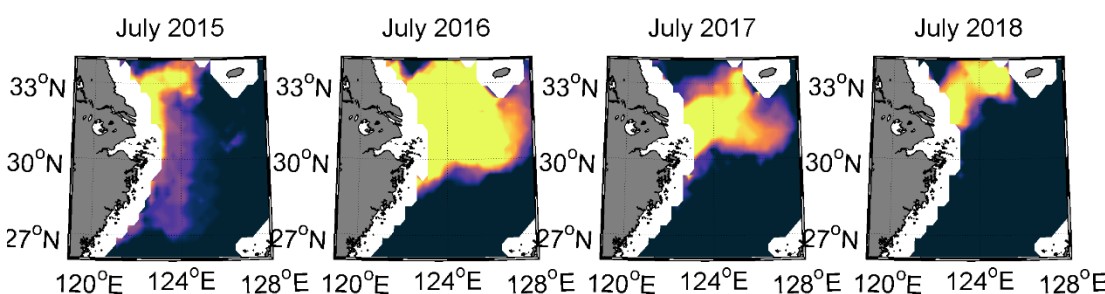

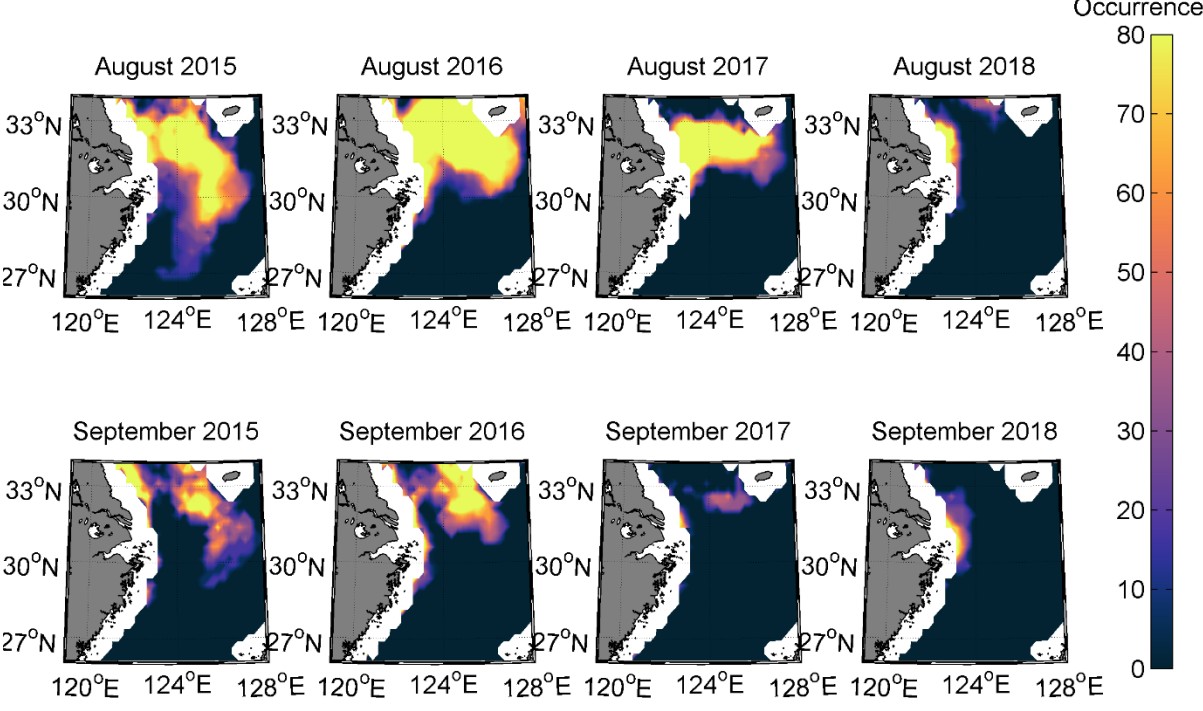

**Figure 11. Maps of monthly occurrence (%) of (remotely sensed) salinity < 30 g kg⁻¹ from June to September, 2015-2018.**


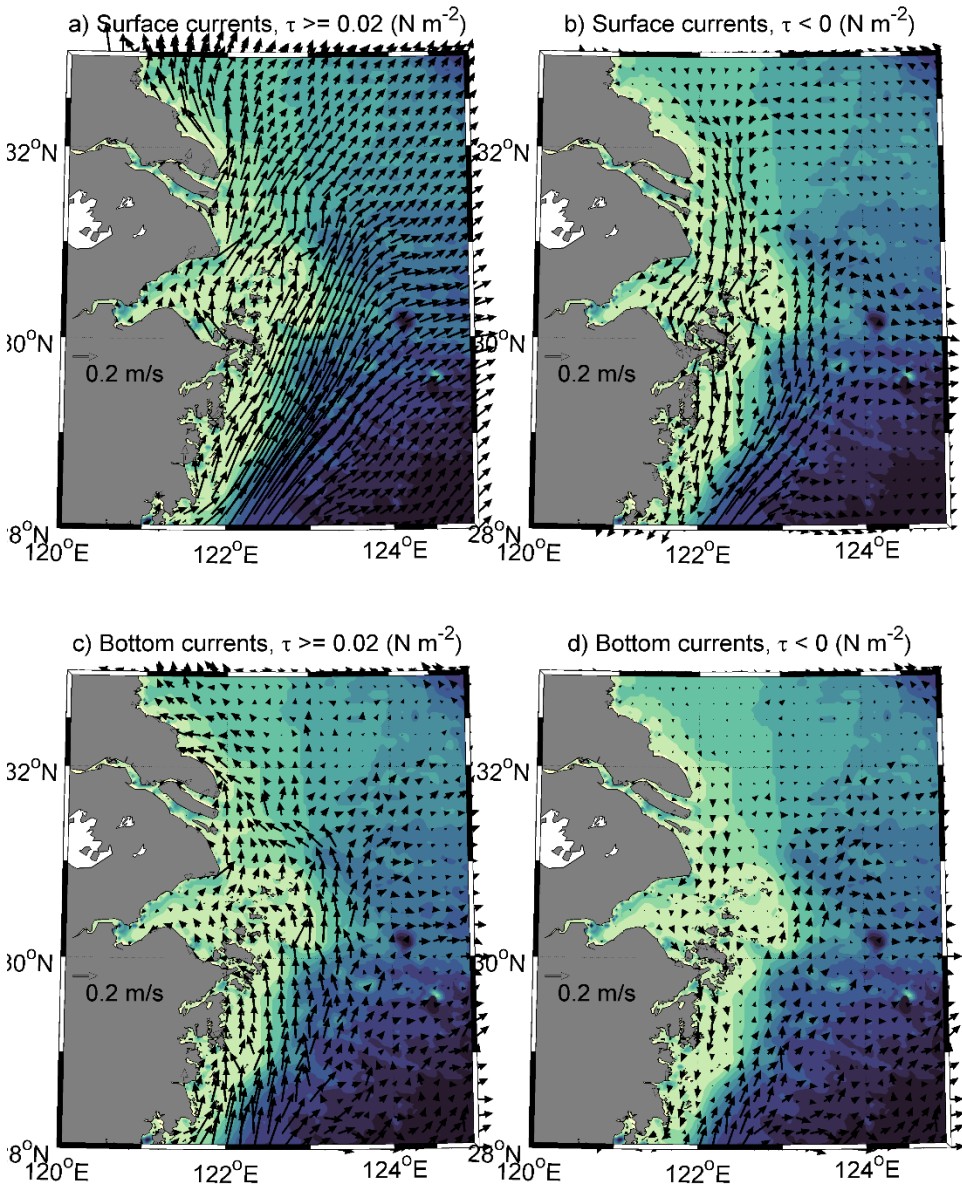

**Figure 12. Maps of surface (upper panels) and near-bottom (lower panels) currents in the sea bottom depths down to 100 m in the cases of wind stress component $\tau_c \geq 0.02$ N m$^{-2}$ (left panels) and $\tau_c < 0$ N m$^{-2}$ (right panels) in 1993-2018. Every fourth current vector is shown.**

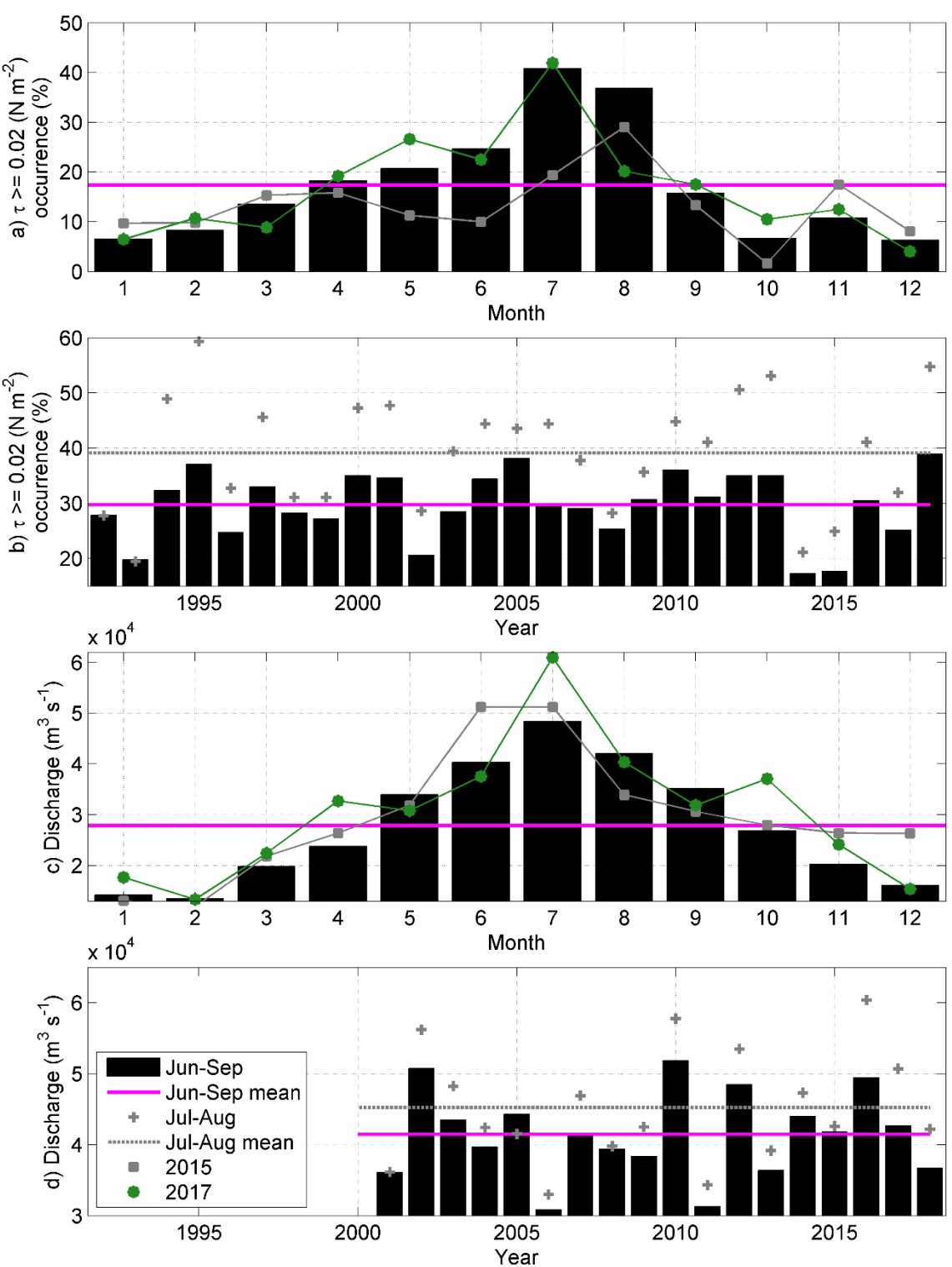


**Figure 13. Annual cycle (a, c) and inter-annual variability (b, d) of wind stress component (1992-2018) $\tau_c \geq 0.02$ N m$^{-2}$ (a, b) and river discharge (c, d) (2001-2018). Blue line shows the overall mean in (a) and (c), and June-September long-term mean in (c) and (d). Red line shows the July-August long-term mean in (b) and (d).**


**Table 1. Summary of general features observed in the study area in summers 2015 and 2017**

| Summer 2015 | Summer 2017 |
|---|---|
| Southward transport of the CDW | East/northeastward transport of the CDW |
| Stronger stratification in the south | Stronger stratification in the north |
| Generally lower SST | Generally higher SST |
| Downwelling along the coast | Upwelling along the coast |
| Higher Chl *a* in south | Higher Chl *a* in north |
| No low DO concentration zone in the north | Strong and shallow hypoxic zone in the north |
| Bottom water warmer and fresher in the north | Bottom water colder and saltier in the north |
| Lower DO concentrations further offshore at sea depths >=30 m in the south | Lower DO concentrations in the coastal upwelling zone in the south |