# Peer review of "Wind-driven stratification patterns and dissolved oxygen depletion off the Changjiang (Yangtze) Estuary"

_Biogeosciences, 2019_

## Referee Comment (RC1) · Fabian Große (Referee) · 17 Nov 2019

**Review of Liblik et al.: "Wind-driven stratification patterns and dissolved oxygen depletion in the area off the Changjiang (Yangtze) Estuary"**

The authors present an analysis of dissolved oxygen (DO) concentrations and hydrography off the Changjiang Estuary observed during two summer surveys in 2015 and 2017. The analysis provides valuable insight in the role of freshwater discharge from the Changjiang and more importantly makes a strong case for the effect of wind forcing on (a) water mass transport/distribution and stratification and (b) coastal upwelling, both affecting the formation of hypoxia in different parts of the region.

As such, this study adds to the evolving knowledge base on hypoxia formation in the East China Sea and is worth considering publication.

However, my major issue with the manuscript at hand is that it is often hard to get what findings are based on what results (due to a lack of figure description and cross-referencing). This leaves me with the impression some of the figures are unnecessary and should be removed. In consequence, the conclusions appear a bit limited and more of a summary of what has been done despite this relatively large number of figures.

Apart from that, the manuscript almost exclusively considers and discusses the physical controls of hypoxia, although it is shown by earlier studies and the one at hand that the physical environment (i.e. stratification, which limits DO supply from oxygenated surface waters to deeper layers) only provides the frame for hypoxia. DO consumption (represented by AOU in this manuscript), driven by primary production and subsequent organic matter degradation, is required for its formation. The described upwelling plays an essential role for enhancing primary production (and subsequent DO consumption), yet this is barely mentioned.

Therefore, I recommend reconsidering the manuscript for publication after major revisions. For specific comments and suggestions please see below.

**General comments**

A large fraction of the figures are not (explicitly) used/described in the Results section, which in parts makes it difficult to follow. In most cases, it is not stated why a specific figure is shown. For instance, what is the purpose of comparing satellite sea surface salinity (SSS) with the mooring time series (Fig. 2), while the mooring data is not used in the rest of the manuscript, nor is satellite SSS? Figures 4 and 5 (transects) are not mentioned explicitly, although they clearly show some important vertical patterns, like the subsurface hypoxic layer reaching almost up to 5m depth (Fig 5). However, this is only mentioned in the discussion. Figures 11-13 are also only mentioned in the Discussion, which is too late. All figures that are relevant for the manuscript need to be described in the Results section in order of their appearance. Irrelevant figures should be removed. It's hard to tell for me, what figures are really important because of the partial lack of description; possibly some panels could be removed from Figs. 3-5.

The AOU figures (although only very briefly discussed) clearly show the important role of DO consumption driven by organic matter degradation for hypoxia formation. However, this factor is only briefly mentioned in both introduction and discussion. Figure 5 shows a distinct increase in AOU in the subsurface from offshore toward the coast. This strong increase indicates that the DO minimum in the near-shore subsurface area is formed via local organic matter remineralisation, although the reduced DO concentrations in the offshore subsurface

waters suggest that the water is preconditioned for hypoxia formation. These things should be discussed as well, as the physics alone cannot explain hypoxia formation. If the survey data contain information on variables that can be used as indicators for organic matter production (e.g. chlorophyll-a/fluorescence observations or nutrient concentrations (as an indicator for enhanced nutrient supply via upwelling), I strongly recommend showing results for the indicator variable(s), e.g. as plots of surface concentrations or vertically resolved transects. This would strengthen the results and discussion, and provide a good connection between physical environment, productivity and DO depletion.

The conclusion reads as if the key findings are (1) that stratification must be present for hypoxia to form (which is nothing new) and (2) that there are two modes of salinity (i.e. stratification) and DO distribution patterns, which are controlled by the prevailing wind field (which is new). The latter is a very interesting finding, however, in terms of conclusions it would be absolutely worthwhile to raise the implications of this finding, e.g. for survey planning, or even for the potential development of a hypoxia forecasting system. Considering that SSS and wind information can be obtained with relatively high spatio-temporal resolution, a combination of both with the findings of the study could be used to make spatially resolved forecasts on likely occurrences of hypoxia, which in turn could be used to optimise hypoxia survey design. The only limitation I see here at the moment is that the connection between wind field, SSS (or Changjiang Diluted Water) and hypoxia is based on only two years, with some additional support from the literature (see Discussion). However, if the authors were able to match up a few more years of high wind stress (according to their study) with corresponding hypoxia patterns, this could at least point into the right direction and provide directions for useful future research. The authors should furthermore discuss what role the different survey timing played for the differences in the observed features as the 2017 survey took place at the beginning of the winter monsoon, while the 2015 one was done in the middle of the summer monsoon. This obviously has a strong effect on the wind field.

Figures: Many figures are not legible in grey-scale, i.e. for colorblind people. I strongly recommend using perceptually uniform color scales, which are available for R, MatLab and Python (and other languages; e.g. https://github.com/matplotlib/cmocean). Color references in figure captions should be avoided for the exact same reason. All figure captions must state clearly what data is shown (e.g. over what period they were averaged etc.)

**Specific comments**
Title: I suggest removing "in the area"

Abstract: The abstract should be rewritten, such that the key messages can be understood without reading the entire manuscript. At this point, it is unclear what the "interaction zone" (line25) between upwelling and surface freshwater is meant to be. High AOU furthermore does not necessarily mean "high DO utilization there" (i.e. local consumption; line 24), especially in the case of advected/upwelled subsurface waters, which are likely to be already undersaturated in DO. Discuss the findings chronologically (first 2015, then 2017).

Introduction:

Lines 50/51: nutrients lead to production, which in turn leads to sinking/sedimentation of organic matter. This is an important aspect, which should also be more emphasized in the discussion (and the results, if possible with the survey data; see my general comment).

Lines 92-94: Again, what about the influence on productivity? Upwelling brings nutrients into the euphotic zone, significantly enhancing organic matter production.

Lines 99-101: I would remove this paragraph.

Data and Methods:
Line 109: There is a 1-month difference in the survey timing between the two years. It would be nice to include a statement in the discussion to what extent relates to the seasonality in the monsoon cycle and if/how it affected the differences between the observations in both years.

Line 110: state the number of stations you used for both years

Lines 119-121: Since you do not describe Fig. 11 (until the discussion), do you need to show/discuss it at all? If not, remove the description of the satellite product and Figs. 2 and 11. If you need it, what is the spatial resolution of the satellite product and how does it match in-situ spatial patterns observed during the survey?

Figure 1: The labels of transects S15/S17 are barely visible. Remove color references in caption.

Lines 122-128: Is the used wind forcing the same that the GLORYS model uses? Please specify. If it's not, I advise using the same one.

Lines 132/133: This is a result and should be stated in the Results section. Where is that shown? Include reference to corresponding figure.

Lines 134-142: Please state why you are analyzing AOU. E.g. to illustrate the role of DO consumption.

Results:
For the entire section: please refer to figures and figure panels where appropriate. Right now it's really hard to know what figure (panel) to look at due to minimal cross-referencing.

Lines 151/152: Please state why you analyse the spatial patterns (e.g. to put the DO observations in context with the physical environment).

Line 154 and Fig. 3: In the figure, you show the 31 isohaline, here you refer to the 30 isoline. Please be consistent. I suggest using the 30 isohaline in the figure.

Lines 164-168: You "describe" 3 figures on 5 lines. It is not possible to understand what finding is based on what figure. It also gives the impression that 1-2 of the figures are superfluous.

Figures 3-5, 6a-2/b-2: Don't use the jet color scale, dark red and dark blue are indistinguishable in grey scale (i.e. for colorblind people). Use standard panel labels, i.e. a, b, c, d etc. (not a-1, a-2, b-1, b-2, …)

Lines 190-192: Upwelling creates favourable conditions for organic matter production (primary production), which then drives DO consumption. You should not skip this process. If you have chlorophyll (or nutrient) data, you should show them to illustrate this.

Line 195: What do you mean with "a certain physical property of water"? Do you mean temperature or salinity or else? Please specify.

Lines 196-199: How did you determine the 2 mg/L AOU-cline to be the "oxycline"? And why do you use the AOU isoline to define your oxycline and not oxygen in the first place? An oxycline is defined by a strong gradient in DO, not by AOU and not by DO depletion (and you are not using a gradient either), so the term cannot be used here. The 2 mg/L AOU isoline further doesn't seem to match the oxycline nor the upper boundary of DO depletion in 2015 (Fig. 4). And what is your basis for using the 24.5°C isotherm to represent the thermocline? A thermocline is also defined by a gradient in temperature, not by a fixed temperature; it doesn't seem to represent the thermocline at transect N15 (Fig. 4). I suggest calculating the pycnocline/oxycline using a gradient approach (e.g. strongest vertical density/DO gradients), which would be much more objective than just picking some isolines. If they still match (which I expect being the case), this would strengthen your statement that the pycnocline determines the upper limit of DO depletion (although this is not really new).

Lines 200-208: I wonder if Fig. 7 adds a lot of information that cannot be obtained from Figs. 4 and 5? The spatial gradients of thermocline depth indicate the strength of upwelling, which can be described using the transects. Same applies to the statement on AOU and the effect of the thickness of the hypoxic layer vs. the DO concentrations. Although I am not sure why this statement is important? In addition, both factors determine AOU and, at transect C15, a thick DO depleted layer coincides with very low DO concentrations, which makes it difficult to quantify what factor is more relevant. I suggest removing this last statement, otherwise the contributions need to be quantified (which does not add to the story).

Lines 243-246: Please provide the equation you use to do this calculation.

Lines 249-254: Here, you possibly could mention the monsoon cycle (in relation to the differences in survey timing) as the 2015 survey took place at the end (beginning) of the summer (winter) monsoon. Then you could also be a more specific with respect to your hypothesis on the main cause for the 2015 vs. 2017 differences.

Lines 256: Does Fig. 8 show wind and currents averaged over the 7-day periods before the surveys or the only averaged over the single day 7 days before the surveys? Please clarify and also clarify in the figure caption. If it's averaged over the single day only, why do you use that one and not the 7-day averages?

Lines 268-270: Is this sentence relevant? Does the negligible bottom current have an important effect on the observed patterns? If yes, clarify. By talking about a buoyant current you also imply that it's baroclinic.

Lines 271-277: Did you calculate the mean winds at the same location as the currents? Please clarify. Further state which different wind directions you used to calculate correlations and to determine the best one. Possibly state the highest values of the other correlations, too, in order to illustrate the difference between the SE-NW direction and the others.

Lines 303-305: I am not sure I understand this statement. You define wind velocity intervals of 0.25 m/s width and average the corresponding current velocities simulated by the model, right? Perhaps you can explain this more clearly. Also, do you do this for every model grid cell in the area of interest or just for the location marked in Fig. 8?

Discussion:
Lines 335-338: You refer to Ekman transport, yet you do not show any analysis of it. You do show near-surface currents (0-5 m depth; Fig. 8), however, the Ekman depth (over which you would need to average/integrate to get Ekman transport) is likely deeper than 5 m and will vary depending on the wind speed. It would indeed be nice if you showed the Ekman velocities in order to illustrate the upwelling conditions.

Lines 339-355: Figures 11-13 have not been mentioned before here; you need to do this in the Results section. It is not possible to follow this discussion without prior description of these results. I also do not see the benefit of discussing 2016 since you do not provide information on the DO conditions in that year. The last sentence in this paragraph is entirely speculative, unless you provide information on the DO conditions in 2016 and 2018 (e.g. from other studies if available).

Lines 360-368: This qualitative comparison of the findings of this study with existing literature is very useful and it suggests that the wind stress (in combination with river discharge) could possibly be used for the development of a simple forecast of hypoxia occurrence using remotely sensed SSS and wind information. This could be discussed, although it may need support by more examples than only the few years mentioned. There is some more literature on hypoxia observations in the East China Sea, which the authors may want to check, e.g.:
Zhu et al. (2011) https://doi.org/10.1016/j.marchem.2011.03.005
Zhu et al. (2017) https://doi.org/10.1016/j.marpolbul.2017.07.029
Li et al. (2002) https://link.springer.com/article/10.1360/02yd9110
If the patterns described in these papers match the "wind-based likelihood" of hypoxia as suggested by this study, the authors could make a case here, which could provide a good direction for future work.

Line 363: "probably occurred" is too speculative. Either you have support for this statement or you should rephrase it, e.g., to "could have occurred"

Lines 377-379: Irrelevant. Remove the whole paragraph

Lines 381/382: The wind-driven near-surface transport offshore is the cause for coastal upwelling of subsurface waters. Please rephrase. The term "upwelling-CDW interaction zone" is not very clear. I understand what you mean, but I would suggest not using this term as it is a bit misleading. The two water masses do not really *interact* with each other, it is rather a displacement of CDW and its replacement by upwelled water.

Lines 386-390: This is one of the few occasions where the role of primary production and nutrient supply is mentioned. This should be expanded and, if possible, strengthened with observations of chlorophyll or phosphate in the Results section.

Lines 396-397: This could be shown more clearly by drawing oxycline and thermocline in Figs. 4 and 5. I think Fig. 7 is unnecessary.

Lines 440-444: None of this analysis is shown/described, so it cannot be discussed. If you want to make a statement on potential future changes (or no changes) due to wind, you need to show a figure comparing future wind projections with current winds. Using only one projection from a single model further doesn't allow for such a strong statement. Please provide references for projected increases in SST and eutrophication if you want to keep the statement.

Conclusions:
The conclusions in general are too weak and rather a summary.
Lines 452-454: Why is the statement on the inclination important?

**Technical corrections**
Line 21: Changjiang Diluted Water (CDW)
Line 26: patterns; are prevailing; summer
Line 29: while the
Line 32: likely the main determinants of the
Line 47: "DO" needs to be introduced here (first time used in main text)
Line 53: below the
Line 55: Hypoxia is a
Line 57: related to eutrophication
Line 62: is characterized by; a steep slop
Line 63: occupy the southern
Line 70: below the
Line 71: the pycnocline
Line 77: are enhanced
Line 79: the Chinese Coastal Current; remove "(TWC)", you don't use it in the manuscript
Line 83: East China Sea
Line 90: hydrographical; in a particular summer
Line 91: in more general context
Line 92: The hypothesis
Line 94: also cause
Line 96: describe
Line 97: during two cruises in the summers of 2015 and 2017; explore
Line 98: investigate

Line 100: related variables

Line 114: for 2015 and 2017 were analyzed

Line 122: "resolution" instead of "grid size"

Line 125, Equation 1: It looks like capital "i" letters in the equation? Should it be the norm of U (i.e. $|U|$)? Please correct.

Line 126: add "above sea level" at end of sentence

Line 128: with approximately 8 km (1/12°) resolution

Line 135: present; apparent oxygen utilization

Lines 135-142: Make sure the font style is used consistently throughout the manuscript for AOU (either italics or normal)

Line 150: Provide a more descriptive title for the subsection, e.g. "Spatial patterns of hydrography and oxygen"

Line 154: The practical salinity scale is a unitless scale, i.e. don't write "psu" here and throughout the manuscript.

Line 156: the 25 isohaline

Line 158: has been prevailing

Line 160: The sea surface temperature (SST)

Line 161: avoid the reference to the Zhoushan Islands or mark them in Fig. 1, otherwise the reader not familiar with the region does not know where to look

Lines 162/163: "… in both years, but colder waters covered smaller areas in 2017." The other part of the sentence is a repetition.

Line 169: Spatial patterns in stratification are described (you don't show changes). What equation of state did you use to calculate density?

Line 172: in areas of 20-30 m bottom depth; no blank after 124

Line 180: in areas of 25-60 m bottom depth

Line 187: less saline OR low-salinity

Line 194: in regions of different depths

Lines 198/199: State the exact number of profiles used for the calculation (i.e. "n = xyz" and not "n < 44/34")

Line 200: and below DO depletion can develop.

Line 201: of both clines

Line 210: summarized in Table 1.

Line 212: What do you mean with "quantified forcing"? Please clarify.

Line 236: during the week

Line 238: what exactly do you mean with river plume **bulge**? It's not clear to me.

Line 239: B should be italicized in the text

Line 240: Add the unit of f (1/s)

Line 241: in 2015 and 2017, respectively.

Line 243: of the buoyant

Line 249: Several indicators (remove "While")

Line 257: The mean wind direction

Line 259: is clearly visible

Line 279: the near-surface offshore flow

Line 283: according to Eq. (1), your $\tau$ can be negative. In this case, your u* would be a complex number. Please use $|\tau|$ in the equation for u*; use $\rho$ for density not q

Line 286: which occur on shorter time scales probably causing deeper mixing.

Line 305: introduce $w_c$

Line 308: What/where is S1?
Line 327: In the upwelling region
Line 329: remove "As we have argued"
Line 339: In Fig. 11, we …
Line 340: 2015 differs
Line 384: one blank too many before Yang et al
Line 395: hypoxia was terminated
Line 398: published a valuable
Line 399: time series; please change throughout the manuscript
Line 406: Fig. 2; In these cases, DO
Line 409: at the coastal
Line 412: such as
Line 413: spatio-temporal resolution

Figure 1: use different colors for transects S15 and S17
Figs. 3-5 use different colors
Figures 4/5: State in the caption what the crosses in the top panels are. Use same color scale for oxygen as in Fig. 3 (0 to 8). Draw thermocline (or pycnocline) in oxygen and AOU panels. Remove remnants of scale labels on left side of top right panel in Fig. 5.
Fig. 6: add labels to panels stating the shown quantity; change units for density difference to $kg/m^3$; remove last sentence in caption and state that AOU was calculated according to Eq. (3) instead; plot hypoxia outline (i.e. 3 mg DO $L^{-1}$) in all panels; use different colors for DO
Fig. 8: Clarify the averaging period and include the depth ranges for the currents in the caption. Remove last sentence in caption
Fig. 9: remove sentence with histogram; add label to color scale
Fig. 10: use colors that are better distinguishable in grey scale; only show legend in first panel and only use "2015" and "2017" in the legend
Fig. 11: use different colors; red areas are not identifiable as high-frequency areas in grey scale; increase size of color scale and add label
Fig. 12: use thicker arrow lines (like in Fig. 8); include the depth ranges for the currents in the caption.
Fig. 13: Use different line styles and for easy distinguishing of  means in grey scale

---

## Referee Comment (RC2) · Anonymous Referee #2 · 25 Nov 2019

This study addressed two different spatial patterns of hypoxia off the Changjiang Estuary (CE) and discussed the impact of the CDW and wind on hypoxia distributions. But I have two main concerns.

1. Observations from 27 August to 5 September in 2015 and from 24 to 29 July in 2017, are actually snapshots in different months and different years. So, their differences result from seasonal or interannual variation or both? In fact, the hypoxia off the CE has prominent seasonal variation and annual cycle, e.g., hypoxia appearing in coastal areas south of the CE in early summer and severe hypoxia in the area north of the CE in August, and also interannual variations (Zhu et al., 2011). The conclusion mentioned

that the annual cycle was dominated by wind and the interannual variation by wind and river runoff. But the manuscript did not provide enough evidence for these conclusions.

2. The CE and adjacent area are highly dynamic and complicated, affected by the river plume, the Yellow Sea Coastal Current, East China Sea Coastal Current, the Taiwan Warm Current and the Kuroshio. The influence of the intrusion of the TWC and Kuroshio on hypoxia has been discussed previously. But the manuscript just considered the river runoff and wind, and did not discuss the role of the TWC and Kuroshio. The intrusion can be recognized by the current pattern, bottom salinity and temperature. The authors should analyze the differences of open ocean intrusion in 2015 and 2017 and their impact on hypoxia distributions.

Specific comments: Line170: how to define weak or strong stratification based on density difference here? Line239: U is the river velocity. How is U calculated?

---

## Author Comment (AC1) · 30 Dec 2019

Dear Anonymous reviewer,

Thank you for reviewing the paper. We have taken both of your main concerns seriously and have made corrections accordingly. Please find our specific responses below. Please notice that our replies include preliminary texts from the revised manuscript. We might do small changes in language, in this phase revised manuscript won't be submitted yet.

Best wishes, Taavi Liblik

[Figure]

Comment: 1. Observations from 27 August to 5 September in 2015 and from 24 to 29 July in 2017, are actually snapshots in different months and different years. So, their differences result from seasonal or interannual variation or both? In fact, the hypoxia off the CE has prominent seasonal variation and annual cycle, e.g., hypoxia appearing in coastal areas south of the CE in early summer and severe hypoxia in the area north of the CE in August, and also interannual variations (Zhu et al., 2011). 2. The conclusion mentioned that the annual cycle was dominated by wind and the interannual variation by wind and river runoff. But the manuscript did not provide enough evidence for these conclusions.

Reply: The annual cycle and inter-annual variability were discussed in the first two sections of the discussion in originally submitted manuscript. In the revised manuscript we have dealt with this question more deeply. Since the 2015 measurements are done one month later in the end of summer, probability of the occurrence of northerly wind is much higher. This is clear from the Fig. 13a. Therefor the snapshot of 2015 describes rather late summer situation and 2017 rather the mid-summer situation indeed. For the inter-annual variability we have analyzed wind and river data (1993-2018) and compared later years with remotely sensed salinity (2015-2018). Mentioned data well agrees with the concept we are suggesting. 2015 distinguishes as special summer in terms of wind forcing and CDW. Weaker summer monsoon resulted in more frequent CDW spreading to the south (as we also observed). Summer 2017 was closer to the "typical" climatological mean summer. We have checked also earlier studies (added more examples during revision, including Zhu et al. 2011) and found similar observations as you mentioned in August. If summer monsoon is at least close to climatological mean, then hypoxia in north is typical. Shortly, our observations somewhat reflect the annual cycle, but 2015 was also special year with weak summer monsoon. We have added also citing to other similar years, when hypoxia in south was observed (Wang and Wang, 2007, Li et al. 1999). The statement in the conclusion you referred was not well worded. We cannot say "Wind forcing and river runoff are main contributors", but we believe we can say based on the study: river runoff and wind are important (not

saying it is main) factors in inter-annual and annual cycle of DO.

Action: We added following text to the second section of discussion: " Thus, our observations conducted in late August - early September 2015 and late July 2017 illustrate the annual cycle of forcing and latter reflect in oxygen and stratification patterns. On the other hand, summer monsoon and river discharge were close to average in summer 2017 while the summer monsoon was clearly weaker in 2015 (Fig. 13a-b). Thus, our observations reflect also the differences in forcing and concurrently in DO distributions between two summers" and to the fourth section: "However, on the top of the inter-annual variability and annual cycle are synoptic scale changes of wind-driven currents and river forcing, which influence the distributions (Fig. 10). Thus, when planning future hypoxia related measurement campaigns, it is worthwhile to take into account wind-driven transport, river discharge and remotely sensed salinity to forecast spreading of the CDW and potential hypoxic area location prior field works. This could allow more efficient use of ship time and more detailed sampling of the hypoxic area."

Statement you referred is modified: "Wind forcing and river runoff are important contributors of inter-annual variations and annual cycle, determining the size and location of low DO areas. The DO minimum is located more likely in the northern part in July-August during summer monsoon and in the southern part during rest of the stratified period."

Comment: 2. The CE and adjacent area are highly dynamic and complicated, affected by the river plume, the Yellow Sea Coastal Current, East China Sea Coastal Current, the Taiwan Warm Current and the Kuroshio. The influence of the intrusion of the TWC and Kuroshio on hypoxia has been discussed previously. But the manuscript just considered the river runoff and wind, and did not discuss the role of the TWC and Kuroshio. The intrusion can be recognized by the current pattern, bottom salinity and temperature. The authors should analyze the differences of open ocean intrusion in 2015 and 2017 and their impact on hypoxia distributions.

Reply: Thank you for this comment. We think this suggestion helped to improve the manuscript. Indeed the role of intrusion was hidden in the manuscript, although we are well aware of intrusion of subthermocline water.

Action: We have made changes thorough manuscript to highlight importance of the intrusion. We defined the upper boundary of the Kuroshio subsurface water mass (as isoline 24.5 temperature) and relate it to the upper boundary of oxygen depletion (2 mgl-1). This simplification is needed as we don't want to go to fine details of sub-surface water mass formation in this study. The same isolines were used also in the original manuscript, but named as thermocline and oxycline. We believe the terms we use now, are more correct. Our main suggestion is that position of intrusion is largely related to the wind forcing. Intrusion climbs higher to coastal slope, as compensation for the offshore transport in the surface layer. And it is located deeper and further offshore, when there is downwelling with northerly winds. Our second suggestion is that existence of intrusion is necessary precondition for hypoxia formation. In order to come up with the two suggestions and react to your comment we: (1) described and analyzed both boundaries more deeply in subchapter 3.1. Particularly, the vertical sections are described more comprehensively in the revised version. 2015 and 2017 are compared in the context of intrusion there. (2) We have added satellite altimetry to give more evidence to our suggestions. The two surveys differ each other clearly by upwelling (2017) and downwelling (2015) and we claim the main reason for this dif-ference is wind forcing. (3) We have highlighted the importance of intrusion, related to oxygen depletion in discussion: "Importance of KSSW thickness on the oxygen de-pletion estimations reveal well also if near bottom oxygen maps are compared with the total AOU maps (Figs. 6g-h). Bottom hypoxia in north in 2017 was much more intense comparing to hypoxia in south in 2015. However, the total AOU was similar in hypoxic zones in both years due to thicker oxygen depleted layer, i.e. thicker KSSW in south in 2015." (4) We have added new section to the discussion highlighting the impor-tance of intrusion. The main point we make there is that strong stratification strength as such, does not necessarily mean hypoxia. We copy the last part of the section

here." Two features must be present for hypoxia formation: 1) KSSW, 2) CDW and/or subsurface water upwelling. We can conclude that colder KSSW determines where (including in what depths) hypoxia could develop. Thus, latter provides necessary precondition for hypoxia. The CDW spreading and/or surbsurface water upwelling (and related biogeochemical, biological processes) determine the magnitude, exact location and timing of oxygen depletion. (5) Since in this paper we deal with the position of the intrusion only, we added another section to discussion that mentions other aspects of intrusion: "Besides the barrier effect by creation of the thermocline, intrusion of KSSW has other implications on oxygen dynamics. First, the subsurface water is oxygen depleted already before local impact of oxygen consumption. The furthest stations in the southeast (Fig. 1) had AOU of 2-2.5 mg l-1 in the deep layer in 2017, i.e. in the same order that has been estimated in the KSSW before (Qian et al., 2017). The total AOU in the water column there was 50-60 g m-2 (Fig. 6h). This water is still rather well ventilated comparing to the deep layer waters that had been impacted by upwelling or CDW induced production in the surface layer (Fig. 6g-h). Despite its initial oxygen depletion, Kuroshio intrusion is important source of oxygen import to the study area (Zuo et al., 2019). Without this lateral oxygen advection, hypoxia could form much faster in larger area (Zuo et al., 2019). Kuroshio intrusion is nutrient rich (Zhang et al., 2007b; Zhou et al., 2019) and its upwelling or vertical mixing could intensify sequence of primary production in the surface, consequently organic matter sinking producing oxygen consumption in the near-bottom layers.."

Specific comments: Line170: how to define weak or strong stratification based on density difference here?

Reply: we do not define it, just comparing different regions.

Action: We changed "relatively weak" to "weaker" to avoid confusion.

Line239: U is the river velocity. How is U calculated?

Reply: It was calculated as discharge divided by cross-sectional area in the river mouth

(which is a bit subjective, depending on exact location). However, we realized during revision that the calculation and topic is not necessary to mention in results, as there are more sophisticated studies already available, where estimates can be taken.

Action: We now have about this topic section only in discussion: "The faith of the river plume can be separated to the regions and processes: circulating bulge near the mouth and downstream current along the coast (Fong and Geyer, 2002; Horner-Devine, 2009). The question is how much of riverine water remains in the river plume bulge and how much is advected to the neighboring areas. It has been estimated that about 80-90% of the discharge accounts to freshwater transport of coastal current (Li and Rong, 2012; Wu et al., 2013)
* * *

---

## Author Comment (AC2) · 30 Dec 2019

Reply: Thank you for your careful reading and comprehensive review! Manuscript has been improved thanks for your suggestions/comments. We have taken into account most of your comments in revising the manuscript. Please notice that in the replies we provide texts from the draft of revised paper. There might be minor changes (related to language) in the final revised paper submission.

Comment: However, my major issue with the manuscript at hand is that it is often hard to get what findings are based on what results (due to a lack of figure description and cross-referencing). This leaves me with the impression some of the figures are unnecessary and should be removed. In consequence, the conclusions appear a bit limited and more of a summary of what has been done despite this relatively large number of figures.
Apart from that, the manuscript almost exclusively considers and discusses the physical controls of hypoxia, although it is shown by earlier studies and the one at hand that the physical environment (i.e. stratification, which limits DO supply from oxygenated surface waters to deeper layers) only provides the frame for hypoxia. DO consumption (represented by AOU in this manuscript), driven by primary production and subsequent organic matter degradation, is required for its formation. The described upwelling plays an essential role for enhancing primary production (and subsequent DO consumption), yet this is barely mentioned.
Therefore, I recommend reconsidering the manuscript for publication after major revisions. For specific comments and suggestions please see below.

Reply: We agree with the critics on referencing and description of figures and we have dealt with the issue.
We also agree conclusions needed rewriting.
Yes, this is true, we focus on physical processes impacting hypoxia. We believe, the physical environment and its sensitivity to forcing is extremely important for hypoxia formation here. Stratification is just one, but not only aspect. Advection of the diluted water is driven by physical processes, as well is deep water intrusion. Both are required for the existence of hypoxia and both depending on wind forcing. Of course oxygen consumption is required to have hypoxia, but advection and diffusion are important as well in oxygen budget. Likewise, it is important, where all this oxygen consumption driven by primary production happens. Latter is driven by physical forcing. We agree DO consumption and related processes must be more highlighted in this paper. But the focus of the paper stays as it was.

Action: In the revised manuscript citing and description of figures is much more comprehensive. Fixing has been done in the entire manuscript, particularly in the results section. Figures were cited 53 times in the original manuscript, and now reaching 152 times in the revised document.
We have rewritten conclusions according to your specific comments.

We have improved introduction to show the focus of the paper more clearly. Primary production and consequent organic matter in the context of consumption is now much more highlighted in the manuscript, particularly in introduction and discussion. Likewise we mention this now in results and conclusions. We have added chl-a maps to give background information about the initial causes of consumption, but we keep our focus on wind forcing and related effects on oxygen fields. Otherwise we rely on existing papers, which have dealt with biogeochemical processes and have shown well the importance of the CDW and upwelling on oxygen consumption in this region.

**General comments**

Comment: A large fraction of the figures are not (explicitly) used/described in the Results section, which in parts makes it difficult to follow. In most cases, it is not stated why a specific figure is shown. For instance, what is the purpose of comparing satellite sea surface salinity (SSS) with the mooring time series (Fig. 2), while the mooring data is not used in the rest of the manuscript, nor is satellite SSS? Figures 4 and 5 (transects) are not mentioned explicitly, although they clearly show some important vertical patterns, like the subsurface hypoxic layer reaching almost up to 5m depth (Fig 5). However, this is only mentioned in the discussion. Figures 11-13 are also only mentioned in the Discussion, which is too late. All figures that are relevant for the manuscript need to be described in the Results section in order of their appearance.
Irrelevant figures should be removed. It's hard to tell for me, what figures are really important because of the partial lack of description; possibly some panels could be removed from Figs. 3-5.

Reply: We agree the critics about description of figures and results. Sea surface salinity was used both in results and discussion part of the manuscript (Fig. 11). We agree, figures 3-5 were not described well enough in detail in the original manuscript. All figures except figure 13 are mentioned in the results.

Action: We have extended the manuscript to give much more details about these figures, and each subplots have been separately cited in the context.
We also considered the suggestion to include figure 13 in results part, but its major content is very related to earlier studies. Thus, we think it is better to start using it in discussion. Otherwise citing to figures is more extensive in the revised manuscript.

Comment: The AOU figures (although only very briefly discussed) clearly show the important role of DO consumption driven by organic matter degradation for hypoxia formation. However, this factor is only briefly mentioned in both introduction and discussion. Figure 5 shows a distinct increase in AOU in the subsurface from offshore toward the coast. This strong increase indicates that the DO minimum in the near-shore subsurface area is formed via local organic matter remineralisation, although the reduced DO concentrations in the offshore subsurface waters suggest that the water is preconditioned for hypoxia formation. These things should be discussed as well, as the physics alone cannot explain hypoxia formation. If the survey data contain information on variables that can be used as indicators for organic matter production (e.g. chlorophyll-a/fluorescence observations or nutrient concentrations (as an indicator for enhanced nutrient supply via upwelling), I strongly recommend showing results for the indicator variable(s), e.g. as plots of surface concentrations or vertically resolved transects.
This would strengthen the results and discussion, and provide a good connection between physical environment, productivity and DO depletion.

Reply: We agree, the background of the do consumption was not well enough mentioned in the previous version of manuscript.

Action: Features in vertical sections are described much more detailed in the revised manuscript. Likewise, information related to consumption is more highlighted in introduction, results, discussion and conclusions. We have added chl-a to the manuscript (to fig. 6) and described it. However, we believe that consumption related processes are quite intensively studied and going deeper into that topic in this paper is out of focus (as wind forcing has been out of focus in many other studies, which have dealt with link between primary production and hypoxia). The preconditioning of offshore water aspect is now included in the discussion.

Comment: The conclusion reads as if the key findings are (1) that stratification must be present for hypoxia to form (which is nothing new) and (2) that there are two modes of salinity (i.e. stratification) and DO distribution patterns, which are controlled by the prevailing wind field (which is new). The latter is a very interesting finding, however, in terms of conclusions it would be absolutely worthwhile to raise the implications of this finding, e.g. for survey planning, or even for the potential development of a hypoxia forecasting system.

Reply: Thank you for the good suggestion.
Action: the first point about stratification, this is much more specific in the revised manuscript (mainly because of reviewer 2 comment). It says now "Pycnocline created by Kuroshio subsurface water is precondition and determines where hypoxia could develop". There is much more analysis dealing with this issue thorough results and discussion section in the revised manuscript. Your recommendation for conclusion is included now: "There is a strong connection between the upper boundaries of Kuroshio intrusion and oxygen depletion. The sensibility of the boundaries to wind forcing shapes oxygen conditions considerably in the area. Autonomous measurement campaigns by mooring arrays and underwater gliders could considerably improve the knowledge about related processes. Concepts suggested in the present work can be utilized, when planning in-situ experiments. Wind, river discharge, remotely sensed salinity and altimetry data can be used to forecast hydrographic situation and potentially hypoxic areas prior field works."

Comment: Considering that SSS and wind information can be obtained with relatively high spatio-temporal resolution, a combination of both with the findings of the study could be used to make spatially resolved forecasts on likely occurrences of hypoxia, which in turn could be used to connection between wind field, SSS (or Changjiang Diluted Water) and hypoxia is based on only two years, with some additional support from the literature (see Discussion). However, if the authors were able to match up a few more years of high wind stress (according to their study) with corresponding hypoxia patterns, this could at least point into the right direction and provide directions for useful future research. The authors should furthermore discuss what role the different survey timing played for the differences in the observed features as the 2017 survey took place at the beginning of the winter monsoon, while the 2015 one was done in the middle of the summer monsoon. This obviously has a strong effect on the wind field.

Reply: Thank you for good suggestions.
Action: We have added more literature examples to discussion. And we have added a subsection to the discussion about survey planning.
"Thus, when planning hypoxia related measurement campaigns in future, it is worthwhile to take into account wind-driven transport, river discharge, remotely sensed salinity and altimetry to forecast spreading of the CDW, upwelling occurrence and deep water intrusion

and according to latter factors estimate potential hypoxic area prior to field works. This could allow more efficient use of ship time and more detailed sampling of the hypoxic area. "

Yes, timing is important. In the original manuscript, the first section of discussion was initiated by the timing question. But somehow the most obvious fact (which you mention) we did not say out. It has been made in the revised manuscript. In short, timing matters, but the wind conditions in 2015 were not only related to the beginning of winter monsoon (you have mixed years in your comment), but the whole summer stands out times-series (Fig. 12) as with weak summer monsoon. This reveals also in Fig. 11, where we show the occurrence of CDW (by remotely sensed salinity). As a result hypoxia developed in south. We have added an additional reference, where the surveys were also from July and October in 2015. There was more oxygen depletion in north in July, but not hypoxia and very strong hypoxia in south in October.

We have added "Thus, our observations conducted in late August - early September 2015 and late July 2017 illustrate the annual cycle of forcing and latter reflect in oxygen and stratification patterns. On the other hand summer monsoon and river discharge were close to average in the whole summer of 2017 while the summer monsoon was clearly weaker in 2015 (Fig. 13a-b). Thus, our observations reflect also the differences in forcing and concurrently the DO distributions between two summers."

And

"It is clear from fig. 11 that CDW transport offshore (to northeast) has occurred in all years, including 2015.  However, one can see how year 2015 differs with low value in the inter-annual time series of wind stress of $\tau_c \geq 0.02$ N m$^{-2}$ (Fig. 13a-b) and with considerable southward advection of the CDW (Fig. 11). Monthly mappings of bottom oxygen in 2015 (Li et al., 2018b) does not show significant oxygen depletion in north in any month while deteriorated hypoxia (comparing to our observations) occurred in October."

Comment: Figures: Many figures are not legible in grey-scale, i.e. for colorblind people. I strongly recommend using perceptually uniform color scales, which are available for R, MatLab and Python (and other languages; e.g. https://github.com/matplotlib/cmocean). Color references in figure captions should be avoided for the exact same reason. All figure captions must state clearly what data is shown (e.g. over what period they were averaged etc.)

Reply: Thank you for bringing this up.

Action: We changed figures according to your suggestions.

**Specific comments**

Comment: Title: I suggest removing "in the area"

Reply: Good idea.

Action: Done.

Comment: Abstract: The abstract should be rewritten, such that the key messages can be understood without reading the entire manuscript. At this point, it is unclear what the "interaction zone" (line25) between upwelling and surface freshwater is meant to be. High

AOU furthermore does not necessarily mean "high DO utilization there" (i.e. local consumption; line 24), especially in the case of advected/upwelled subsurface waters, which are likely to be already undersaturated in DO. Discuss the findings chronologically (first 2015, then 2017).

Reply: We agree.

Action: We have re-written the abstract. We realized interaction as such is indeed not analysed and is not under focus of the present study. We removed it from abstract. The abstract has now been written in chronological order.

Comment: Lines 50/51: nutrients lead to production, which in turn leads to sinking/sedimentation of organic matter. This is an important aspect, which should also be more emphasized in the discussion (and the results, if possible with the survey data; see my general comment).

Reply: We agree this is important aspect.

Action: We have emphasized this topic more comprehensively in introduction of the revised version of the manuscript. Likewise, we have extended the topic in the discussion. We have included Chl a data to the results section. We also mention primary production in conclusions in the revised paper.

Comment: Lines 92-94: Again, what about the influence on productivity? Upwelling brings nutrients into the euphotic zone, significantly enhancing organic matter production.

Reply: I am not 100% sure, if I understand your concern about lines 92-94 (there is no upwelling mentioned). Nevertheless, I understand your general concern.

Action: In the revised manuscript, we have described the role of productivity in the CDW and due to upwelling and the consequences to oxygen depletion more thoroughly. Also we have made more clear, what is the focus of our work. We believe after these changes it is understandable for a reader why we write lines 92-94 like this.

Comment: Lines 99-101: I would remove this paragraph.

Reply: We removed.

Comment: Line 109: There is a 1-month difference in the survey timing between the two years. It would be nice to include a statement in the discussion to what extent relates to the seasonality in the monsoon cycle and if/how it affected the differences between the observations in both years.

Reply: Yes, we agree. This topic was dealt in the first section of the discussion (original manuscript) but clear statement about the matter was missing.

Action: We extended there: "Both, maximum frequency of southerly wind and river discharge in the annual cycle occur in July-August (Figs. 13a,c). Thus, our observations conducted in

late August - early September 2015 and late July 2017 illustrate the annual cycle of forcing and latter reflect in oxygen and stratification patterns."
But we also need add: "On the other hand summer monsoon and river discharge were close to average in the whole summer of 2017 while the summer monsoon was clearly weaker in 2015 (Fig. 13a-b). Thus, our observations reflect also the differences in forcing and concurrently the DO distributions between two summers. "

Comment: Line 110: state the number of stations you used for both years

Reply and action: We added (65 in 2015 and 49 in 2017).

Comment: Lines 119-121: Since you do not describe Fig. 11 (until the discussion), do you need to show/discuss it at all? If not, remove the description of the satellite product and Figs. 2 and 11. If you need it, what is the spatial resolution of the satellite product and how does it match in-situ spatial patterns observed during the survey?

Reply: We checked again and found that we have described and cited Fig. 11 under the results section, please see 295-302 (original submission). Resolution is 0.25°.

Action: We need to show and discuss figure 11. We considered merging of the Fig. 11 related text from discussion to results. But we realized this text (though have some results) has many discussion elements. So we prefer to keep it there. However, we added text after the section we just mentioned (295-302, original submission). We introduce there also 2016/18 sea surface salinity observations (CDW spreading).
We added information about resolution to data and methods section. Match between satellite and in-situ salinity is quite good. We don't want to add another figure just for comparison. Time-series comparison (Fig. 2) shows well that remotely sensed salinity is useful product in this region. We think fig. 2 is a good indicator of remotely sensed salinity quality in this area also for further studies. There are very limited time-series available in this region.

Comment: Figure 1: The labels of transects S15/S17 are barely visible. Remove color references in caption.

Reply and action: Done

Comment: Lines 122-128: Is the used wind forcing the same that the GLORYS model uses? Please specify. If it's not, I advise using the same one.

Reply: It is not exactly the same. GLORYS uses ERA-Interim wind. For the wind forcing we use the dedicated CMEMS Wind product, which uses remote sensing observations from several satellite missions. However, potential temporal gaps in the observational time-series are filled by ERA-interim. The product has been validated and the quality is well proven the according to the public documents provided by CMEMS.

Action: We prefer to keep wind forcing as it is now.

Comment: Lines 132/133: This is a result and should be stated in the Results section. Where is that shown? Include reference to corresponding figure.

Reply: We agree, it was repeating of results.

Action: We removed it from here. We state this also in results section and cite to figure 8 (simulated currents) and 3 (observed salinity).

Comment: Lines 134-142: Please state why you are analyzing AOU. E.g. to illustrate the role of DO consumption.

Reply and action: We added suggested sentence.

Comment: For the entire section: please refer to figures and figure panels where appropriate. Right now it's really hard to know what figure (panel) to look at due to minimal cross-referencing.

Reply: We agree.

Action: We have increased number of references to figures considerably. We cite to panels.

Comment: Lines 151/152: Please state why you analyse the spatial patterns (e.g. to put the DO observations in context with the physical environment).

Reply: We agree.

Action: We added to the very beginning: "In order to link the thermohaline structure and DO observations we next analyze temperature.."

Comment: Line 154 and Fig. 3: In the figure, you show the 31 isohaline, here you refer to the 30 isoline.
Please be consistent. I suggest using the 30 isohaline in the figure.

Reply: We fixed.

Action: There are 30 both in the figure and text of the revised manuscript.

Comment: Lines 164-168: You "describe" 3 figures on 5 lines. It is not possible to understand what finding is based on what figure. It also gives the impression that 1-2 of the figures are superfluous.

Reply: We agree with critics.

Action: We have solved this issue. Figures are now cited by subplots. Also more text is added to describe the results more comprehensively.

Comment: Figures 3-5, 6a-2/b-2: Don't use the jet color scale, dark red and dark blue are indistinguishable in grey scale (i.e. for colorblind people). Use standard panel labels, i.e. a, b, c, d etc. (not a-1, a-2, b-1, b-2, …)

Reply:  We changed.

Action: We changed colorscale and changed panel labels according to your suggestion.

Comment: Lines 190-192: Upwelling creates favourable conditions for organic matter production (primary production), which then drives DO consumption. You should not skip this process. If you have chlorophyll (or nutrient) data, you should show them to illustrate this.

Reply: We made changes accordingly.

Action: We have added more text on that topic. We have added chl-a to illustrate the process. Revised text is: "This indicates that coupling between coastal upwelling, which bring subsurface water to shallower depths (euphotic zone) and fresher riverine surface water, created favorable conditions for the primary production in the upper layer and concurrent DO consumption and depletion in the deep layer (Figs. 5h,i,k,l). Higher Chl $a$ concentrations in the upwelling areas revealed from the satellite image, although not as high as in north (Fig. 6d)"

Comment: Line 195: What do you mean with "a certain physical property of water"? Do you mean temperature or salinity or else? Please specify.

Reply: We specified.

Action: We added: ", e.g. to a temperature-, salinity- or density isoline."

Comment: Lines 196-199: How did you determine the 2 mg/L AOU-cline to be the "oxycline"? And why do you use the AOU isoline to define your oxycline and not oxygen in the first place? An oxycline is defined by a strong gradient in DO, not by AOU and not by DO depletion (and you are not using a gradient either), so the term cannot be used here. The 2 mg/L AOU isoline further doesn't seem to match the oxycline nor the upper boundary of DO depletion in 2015 (Fig. 4). And what is your basis for using the 24.5°C isotherm to represent the thermocline? A thermocline is also defined by a gradient in temperature, not by a fixed temperature; it doesn't seem to represent the thermocline at transect N15 (Fig. 4). I suggest calculating the pycnocline/oxycline using a gradient approach (e.g. strongest vertical density/DO gradients), which would be much more objective than just picking some isolines. If they still match (which I expect being the case), this would strengthen your statement that the pycnocline determines the upper limit of DO depletion (although this is not really new).

Reply: Yes we agree in that, thermocline and oxycline are not the best terms. The reason why the gradient approach (what you suggest) was not used and why we still don't want to use it, is the fact that if there is upwelling event (like in 2017), then the gradient approach fails. We would like to show surfacing of the both isolines. Gradient approach does not allow that. Likewise, gradient approach has some other problems (e.g. small intrusions, which are not interests of the present study, cause jumps in space). I (T. Liblik) have dealt with this issue on 35 sections measured across the Gulf of Finland (http://www.borenv.net/BER/pdfs/ber22/ber22-027-047-Liblik.pdf). For similar reasons we did not use finally the gradient approach, but used isolines (which varied for every section though) in that study.

Action: We rephrased the terms accordingly, so we don't call the isolines anymore as thermocline and oxycline. We have changed that in the whole manuscript. 24.5° describes the

upper boundary of the colder water mass (Kuroshio intrusion) while the 2 mg/l AOU describes the upper boundary of oxygen depletion. This is done in the whole manuscript. In relation to the comment of other reviewer we now put much more attention to these boundaries in the whole manuscript. It also means that we don't talk about stratification in general way anymore (as in the original manuscript). We have also added altimetry data (mean sea level anomaly) to confirm the upwelling/downwelling pattern and related changes with isolines. We copy here section from the draft of the revised paper to illustrate the approach/conclusion in the revised paper: " Two features must be present for hypoxia formation: 1) KSSW, 2) CDW and/or subsurface water upwelling. We can conclude that colder KSSW determines where (including in what depths) hypoxia could develop. Thus, latter provides necessary precondition for hypoxia. The CDW spreading and/or surbsurface water upwelling (and related biogeochemical, biological processes) determine the magnitude, exact location and timing of oxygen depletion."

Comment: Lines 200-208: I wonder if Fig. 7 adds a lot of information that cannot be obtained from Figs. 4 and 5? The spatial gradients of thermocline depth indicate the strength of upwelling, which
can be described using the transects. Same applies to the statement on AOU and the effect of the thickness of the hypoxic layer vs. the DO concentrations. Although I am not sure why this statement is important? In addition, both factors determine AOU and, at transect C15, a thick DO depleted layer coincides with very low DO concentrations, which makes it difficult to quantify what factor is more relevant. I suggest removing this last statement, otherwise the contributions need to be quantified (which does not add to the story).

Reply: We think figure 7 is necessary. In figures 4-5 we have selected 3 sections (not all the stations) while here we use all the available stations. It gives the better view about lateral isoline distributions, which is hard to visualize just by selected sections. After revision this figure became even more important. It describes the role of Kuroshio subsurface water as precondition for the hypoxia (as suggested by reviewer 2).
The statement is related to the fact that if we compare 2015/2017 south/north bottom oxygen, we get that northern areas was much more oxygen depleted in 2017. However if we compare the total AOU (which describes the oxygen depletion in the whole water column) the two areas/years have similar values. This means bottom oxygen maps, which is the most common way to illustrate oxygen problem in this region, not necessarily describe the oxygen problem (depletion) solely. Other stratification characteristics matter also, not only the strength of the stratification. We think it is needed to highlight this point for further studies, because oxygen depletion (Oxygen Dept) is one of the indicator of eutrophication (e.g. https://www.frontiersin.org/articles/10.3389/fmars.2019.00054/full).
Yes, that is good point about AOU-layer thickness statement. We have made error with the statement.

Action: We have added sentence about this matter to discussion, where we talk about importance of vertical location and movements of pycnocline (line 409, original submission). We rephrased the statement: "It means that high total AOU there (comparing to neighboring areas) was related both to the thick subsurface DO depleted layer and to the higher (comparing to surrounding area) AOU." The point we wanted to make is that thickness also matters. It is often missed, because not full resolution CTD-profiles are used, but bottle values (surface and bottom) instead. PS. This is another reason why want to show fig. 7.

Comment: Lines 243-246: Please provide the equation you use to do this calculation.

Reply: We added.

Action: We have added the equation to data and methods.

Comment: Lines 249-254: Here, you possibly could mention the monsoon cycle (in relation to the differences in survey timing) as the 2015 survey took place at the end (beginning) of the summer (winter) monsoon. Then you could also be a more specific with respect to your hypothesis on the main cause for the 2015 vs. 2017 differences.

Reply: Yes, this can be already mentioned here.

Action: Added "Since our surveys are conducted annually one month apart, the differences between the two surveys might be associated with the annual cycle in wind climate and river discharge. Our hypothesis is that the dominant factors behind the discrepancy of the two surveys are wind forcing and river discharge, and possibly their seasonality."

Comment: Lines 256: Does Fig. 8 show wind and currents averaged over the 7-day periods before the surveys or the only averaged over the single day 7 days before the surveys? Please clarify and also clarify in the figure caption. If it's averaged over the single day only, why do you use that one and not the 7-day averages?

Reply: We use 7-day averages.

Action: We changed the sentence "Mean wind, surface currents and bottom currents during 7 days prior to the surveys are presented in Fig. 8." and figure caption accordingly.

Comment: Lines 268-270: Is this sentence relevant? Does the negligible bottom current have an important effect on the observed patterns? If yes, clarify. By talking about a buoyant current you also imply that it's baroclinic.

Reply: We agree, indeed the sentence can be omitted.

Action: We removed it.

Comment: Lines 271-277: Did you calculate the mean winds at the same location as the currents? Please clarify. Further state which different wind directions you used to calculate correlations and to determine the best one. Possibly state the highest values of the other correlations, too, in order to illustrate the difference between the SE-NW direction and the others.

Reply: Not in the same location. For all the work one location was used for the wind. We calculated all directions by 10 degree step. We believe giving correlation values of other directions is not necessary. It is higher closer to the best direction (near SE-NW) and it is weakest across the best direction (NE-SW) as expected. We have tried also to use wind over larger area when preparing manuscript, but it did not give any advantage.

Action: We added to the text. "(see the location in Fig. 1)".

We added to the text "…from different directions (full circle by 10° steps)."

Comment: Lines 303-305: I am not sure I understand this statement. You define wind velocity intervals of 0.25 m/s width and average the corresponding current velocities simulated by the model, right? Perhaps you can explain this more clearly. Also, do you do this for every model grid cell in the area of interest or just for the location marked in Fig. 8?

Reply: Yes, it is done as you wrote. We do it for the location marked in Fig. 8.

Action: We tried to rephrase "We grouped the simulated $v_m$ to the wind velocity $w_c$ classes with step of 0.25 ms$^{-1}$ and took average of the each group. By doing this, we found relation between mean $w_c$ and $v_m$." We hope it is clearer now.
We added information to the previous sentence to make it clear: Next we make an attempt to quantify the two CDW spreading cases caused by wind forcing through analyzing meridional currents $v_m$ at section S1 (see Fig. 8) and wind data from the period June–September in 1993-2018."

Discussion:
Comment: Lines 335-338: You refer to Ekman transport, yet you do not show any analysis of it. You do show near-surface currents (0-5 m depth; Fig. 8), however, the Ekman depth (over which you would need to average/integrate to get Ekman transport) is likely deeper than 5 m and will vary depending on the wind speed. It would indeed be nice if you showed the Ekman velocities in order to illustrate the upwelling conditions.

Reply: We have made mistake there, what we meant is Ekman surface current.

Action: We fixed the error.

Comment: Lines 339-355: Figures 11-13 have not been mentioned before here; you need to do this in the Results section. It is not possible to follow this discussion without prior description of these results. I also do not see the benefit of discussing 2016 since you do not provide information on the DO conditions in that year. The last sentence in this paragraph is entirely speculative, unless you provide information on the DO conditions in 2016 and 2018 (e.g. from other studies if available).

Reply: This is not true. Content at section 295-302 (original submission file) is based on Fig. 11 and the figure is mentioned there as well. Same is with the figure 12. It was included in the last section of the results.
You are correct about figure 13 – it appears in the discussion first. However, this figure is so much linked to the discussion that it is difficult to handle it before without repeating, which would be annoying for a reader. The annual cycle aspect is now mentioned (as reaction to your other comment) in the results part. So this information is provided now before discussion. We hope discussion is easier to follow after those changes.
The last sentence there is speculative indeed. Previously in the results we have linked wind, discharge, CDW spreading and hypoxia in 2015/2017. In the same discussion section we show that wind and discharge qualitatively explain CDW spreading. Naturally, we should discuss/hint at least, how the oxygen picture look like based on this limited information.

We have information about extensive hypoxia in north in 2016, which very well confirm our suggestion. Unfortunately, it is difficult to cite this observation (no publication or report available yet to our knowledge).

Action: We have extended section in results about Fig. 11 in revision and included some sentences about 2016, 2018 observations there (as reaction to your other specific comment).

The figure 12 is important to show that mean current structure a week before 2015/2017 surveys (Fig. 8) holds truth also when we take historical winds-currents and group them according to our criteria. We have added such explaining sentence about this aspect to the very end of the results.

We made the speculative statement softer by replacing "should" with "could".

Comment: Lines 360-368: This qualitative comparison of the findings of this study with existing literature is very useful and it suggests that the wind stress (in combination with river discharge) could possibly be used for the development of a simple forecast of hypoxia occurrence using remotely sensed SSS and wind information. This could be discussed, although it may need support by more examples than only the few years mentioned. There is some more literature on hypoxia observations in the East China Sea, which the authors may want to check, e.g.:
Zhu et al. (2011) https://doi.org/10.1016/j.marchem.2011.03.005
Zhu et al. (2017) https://doi.org/10.1016/j.marpolbul.2017.07.029
Li et al. (2002) https://link.springer.com/article/10.1360/02yd9110
If the patterns described in these papers match the "wind-based likelihood" of hypoxia as suggested by this study, the authors could make a case here, which could provide a good direction for future work.

Reply: Thank you for the suggestion.

Action: We have added more examples to the suggested section and also to the section before (where we talk about SSS). We also added sentence about synoptic scale variability until the end of the section.
End of the section now looks like this "Such situation has been captured for instance in 1998 (Wang and Wang, 2007) and in 1999 (Li et al., 2002). However, on the top of the inter-annual variability and annual cycle are synoptic scale changes of wind-driven currents and river forcing, which likely influence the distributions on shorter time scales (Fig. 10). Thus, when planning hypoxia related measurement campaigns in future, it is worthwhile to take into account wind-driven transport, river discharge, remotely sensed salinity and altimetry to forecast spreading of the CDW, upwelling occurrence and deep water intrusion and according to latter factors estimate potential hypoxic area prior to field works. This could allow more efficient use of ship time and more detailed sampling of the hypoxic area. "

Comment: Line 363: "probably occurred" is too speculative. Either you have support for this statement or you should rephrase it, e.g., to "could have occurred"

Reply: We agree.

Action: We rephrased as you suggested.

Comment: Lines 377-379: Irrelevant. Remove the whole paragraph

Reply: The point we wanted to make here is that in the case of southerly winds CDW might impact surrounding ocean also out from our study area. We believe statement hinting that should stay in manuscript.

Action: We rephrased the section. We hope its importance comes out better now. "Offshore, east- or northeastward advected CDW caused by southerly wind, as we observed in 2017, might form detached eddies due to interaction of the Ekman flow and density driven frontal currents (Xuan et al., 2012). Those eddies bring CDW further offshore and alter physical, chemical characteristics (including oxygen conditions) and primary production in the water column (Wei et al., 2017). On the other hand we noted ventilating impact of colder cyclonic eddy in north in 2015.

Comment: Lines 381/382: The wind-driven near-surface transport offshore is the cause for coastal upwelling of subsurface waters. Please rephrase. The term "upwelling-CDW interaction zone" is not very clear. I understand what you mean, but I would suggest not using this term as it is a bit misleading. The two water masses do not really interact with each other, it is rather a displacement of CDW and its replacement by upwelled water.

Reply: Studies in similar environments show that it is not just displacement, but also mixing between two water masses occur (subsurface water and former surface water). E.g. https://www.sciencedirect.com/science/article/pii/S0278434309002064 found that the share of former surface was 15% in the formed mixture after upwelling. But we agree, some other term than interaction can be used here.

Action: We rephrased"inter-action,, with "coupling,,and added the reference of Wei et al. 2017 who have described the"coupling,, more thoroughly. It now says: "Shoreward, upslope penetration of the sub-thermocline KSSW and hypoxia in the upwelling – CDW coupling zone (Wei et al., 2017a) were observed."

Comment: Lines 386-390: This is one of the few occasions where the role of primary production and nutrient supply is mentioned. This should be expanded and, if possible, strengthened with observations of chlorophyll or phosphate in the Results section.

Reply: We agree.

Action: Yes we have expanded the topic and have included chl a for illustration.

Comment: Lines 396-397: This could be shown more clearly by drawing oxycline and thermocline in Figs. 4 and 5. I think Fig. 7 is unnecessary.

Reply: Lines can be added, but we would like to keep fig. 7 (we have explained reasons in other answer).

Action: We added the lines to the figures 4-5

Comment: Lines 440-444: None of this analysis is shown/described, so it cannot be discussed. If you want to make a statement on potential future changes (or no changes) due to wind, you need to show a figure comparing future wind projections with current winds. Using only one projection from a single model further doesn't allow for such a strong statement. Please provide references for projected increases in SST and eutrophication if you want to keep the statement.

Reply: We agree

Action: We removed the section.

Comment: The conclusions in general are too weak and rather a summary.
Lines 452-454: Why is the statement on the inclination important?

Reply: Inclination is one reason why northern and southern hypoxia are so different. Deeper clines in south provide more stable and long lasting conditions for hypoxia.

Action: We have rewritten conclusions. Inclination as such is not mentioned anymore in conclusions.

**Technical corrections**
Thank you again for the careful look! We have made all the changes you suggested.
Action related your comments is "done" or "solved".

Regarding your comment:
Line 238: what exactly do you mean with river plume **bulge**? It's not clear to me.

Reply: We moved this part to discussion, because the topic is investigated more thoroughly by other papers, no need to repeat.

Action: As reaction to your comment we put new sentence there to the beginning: "The faith of the river plume can be separated to the regions and processes: circulating bulge near the mouth and downstream current along the coast (Fong and Geyer, 2002; Horner-Devine, 2009). The question is how much of riverine water remains in the river plume bulge and how much is advected to the neighboring areas. It has been estimated that about 80-90% of the discharge accounts to freshwater transport of coastal current (Li and Rong, 2012; Wu et al., 2013)."

---

## Referee Report (RR1)

**Review of Liblik et al.: "Wind-driven stratification patterns and dissolved oxygen depletion off the Changjiang (Yangtze) Estuary"**

First, I'd like to express that I appreciate that the authors addressed most of my comments on the previous version of the manuscript. In my opinion, the manuscript has improved quite a bit, although I still see some room for improvement.

In particular, I suggest some re-ordering (and shortening) of parts of the Results section and some clarification and streamlining in the Discussion. In addition, some of the key results that are described in the text could be supported by matching figures to make them easier understandable.

Therefore, I recommend reconsidering the manuscript for publication after moderate revisions. For specific comments and suggestions please see below.

**General comments**

The entire section 3.1 is very long and provides a lot of details, which at least partly seem not useful to me. In fact, it rather complicates getting the key information relevant for the story from that section. The back and forth jumping between Figs. 3-7 in the text block from line 190 to line 261 further complicates that. I strongly recommend rearranging that section and possibly the corresponding figures, ideally by introducing a "chronological" order of figures and figure referencing (e.g. all maps first, then the section plots). This would make this part much easier to follow.

I really like the analysis described on lines 411-434 as it establishes the strong link between wind forcing and hydrodynamics, and thus the potential for hypoxia in the northern and southern regions. However, as it's only described by text, it is not delivered as strikingly as it could be. A figure could really help improve that. Possibly, a figure showing exactly these grouped velocities (described in the text) plotted against each other could do the trick?

The discussion could be tightened a bit more by more clearly emphasizing the main aspects found in the study. Similar to the aforementioned Results section, it seems to contain a lot of details among which the key links get a bit buried (especially lines 465-530). In consequence, I still have troubles lining up the essential cause-effect chain from the Discussion. Based on existing literature and the present study, it appears clear to me that stratification and primary production together with the bottom depth (which controls the initial subsurface DO inventory) are the essential factors for the formation of hypoxia off the Changjiang. Stratification is controlled by vertical density gradients, which can form either due to fresh CDW near the surface or onshore transport of cold and saline KSSW or a combination of both. Production is controlled by nutrient availability (during spring to fall), which can be provided either from the CDW or the KSSW via upwelling, whose distribution and occurrence are controlled primarily by the wind forcing. Both is nicely shown in this study, but it's not clearly discussed in this causal context The latter point is confirmed by the discussion of the wind-current link in the context of literature, which could provide the basis for a relatively simple hypoxia forecast.

The discussion part on years with wind conditions supporting hypoxia in the northern and southern regions, respectively, based on the analysis of wind forcing and upwelling/downwelling conditions could also be summarized in a table to make it more convincing (and easier to follow). Instead of listing all the years with their different down-

/upwelling characteristics and the potentially matching hypoxia observations (lines 489-504), a table could be provided including, e.g. year, down-/upwelling favorable, hypoxia in north/south, matching observations (with references).

The text descriptions are sometimes a bit complicated and stilted (but generally understandable). Perhaps you can have a colleague (ideally an English native speaker) have a look at the manuscript to make it more concise and accessible?

**Specific comments**

L78/79: Isn't it DO consumption and vertical supply (via mixing), which are determined by CDW? DO depletion is a result of the two factors.

L97-99: I think the hypothesis is a bit weakly phrased. "Sensitive" suggests that the wind will do something to hypoxia, but your study shows that wind is a key control for the physical environment supporting hypoxia formation. Please rephrase.

L177: Please specify why you chose the 2 mg/L AOU isoline otherwise it appears somewhat arbitrary.

L194/195: Wouldn't AOU (which you also show) be a better indicator for DO consumption? Chl *a* is acceptable as a proxy for production (although it comes with some limitations).

L204/205: This sentence seems to contradict line 194

L196-261: Please describe figures (and panels) chronologically.

L311: River discharge is a buoyancy flux, yes. But is it that buoyancy flux that causes the southward flow or is it the barotropic pressure gradient (imposed by the river discharge) that causes the geostrophic southward flow?

L315-334: I would start this paragraph by stating that two factors can be considered to affect the distribution of CDW in the region: Changjiang River discharge and wind forcing.

L333/334: Given the different survey timing, it is very likely that seasonality/annual cycle of the wind forcing is important. I wouldn't phrase that as a hypothesis.

L364: Perhaps you can start a subsection here as the following parts provide an analysis on larger time scales (i.e. not only based on the two surveys).

L364-370: My understanding of this paragraph is that you calculate cross-correlations for different time lags (shifted by one day each) and wind in the different directions? Is that correct? Please clarify in the text what you did exactly and which lags you tested.

L398: You previously highlight the role of wind direction, while you refer to wind stress here. How do the periods of low wind stress relate to wind direction? Based on Eq. (1), my understanding is that low positive stress means weak northward winds; it could be useful stating that here to make the description less abstract.

L412: I don't think "quantify" is the appropriate term here. Isn't it rather an extrapolation of your survey findings to a longer time period? Or in other words, you're putting the survey observations in a broader context.

L420: Technically, the current is already "altered" if it's slowed down (or sped up). Maybe rephrase it such that you 'define' $v_m = 0$ as current 'alteration' and that you use this term accordingly hereafter.

L433/434: This is a key message that should be made more clearly, like: "This supports our hypothesis that the wind field is a key control for the direction of the Chinese Coastal Current controlling the CDW distribution off the Changjiang River Estuary, and thus hypoxia formation."

L439-444: Perhaps you could start the discussion with a statement that the balance between oxygen consumption in and supply to the subsurface layer controls the formation of hypoxia, and that consumption is controlled primarily by primary production and subsequent organic matter degradation, while supply depends on vertical stratification and lateral transport. Then you could state that your study focuses on the effect of the wind field these supply pathways.

L445-504: I think the general logic of this part works well, however, I recommend tightening it a little bit to make it more concise and easier to follow. State what the key patterns from the two cruises are and what they suggest with respect to the role of the physical environment for hypoxia formation (and to some level productivity, but you don't need to put as much emphasis on that as it's not the main focus). Then discuss the findings from your wind analysis and how the wind field controls the physical environment in the northern and southern regions (and thus the likelihood of hypoxia in both regions). Then you put it in context with observed hypoxia to support your findings.

L456-458: I do not agree with this statement. Yes, the wind conditions were different during the two summers, nevertheless, your observations were at too very different stages of the summer season. Thus, they cannot be compared in this way and they do not reflect inter-annual differences but seasonal differences.

L463: I think the role of the surface current in distributing the CDW (Fig. 11) is the essential ingredient, and should be stated here. The strong haline stratification limits the vertical DO supply in the first place. In case of a deep KSSW intrusion, the additional thermal stratification then defines the potential vertical extent of hypoxia.

L509-510: I don't understand this statement. If river discharge accounts for 80-90% of the coastal current, why does that mean that most discharge does not remain in the river plume? Shouldn't it be the exact opposite as the coastal current determines the distribution of the river plume?

L561-562: I am not fully convinced that it is KSSW along the N17 section (Fig. 5) as temperature and salinity of the subsurface waters along this section (Fig. 5) differ from the other sections. Could it also be water originating from the Taiwan Warm Current? Maybe you could state earlier in the manuscript (e.g. on pages 3 and 5) what the physical properties (T and S) of both water masses are to make clear that it can only be KSSW (if that's the case).

L578-580: Same as the previous comment with respect to KSSW.

L604/605: Why does this mixture promote organic matter settling? Or do you mean primary production (since you mention nutrient consumption) and subsequent settling?

**Technical corrections**
L24: the pycnocline
L26: the manuscript already uses a bunch of acronyms, I would avoid using CCC (throughout the manuscript)
L28/29: a well ventilated area in the north and a hypoxic area; the CCC; reversal of the
L31: offshore transport; a subsurface intrusion
L32: shallow areas (<10 m depth) at the continental slope
L33: in the north
L64: In the literature, hypoxia off …
L65: divided into; state where the division between northern and southern region is usually done (around 30°N); features a shallow
L66: bottom, while

L69: The region is strongly

L74: by a shallow

L77: in consequence

L81: compared to that of wind stirring

L86: are further influenced by

L87: no need for CCC and TWC (rarely used); n comma after (CCC); , which originates

L88: at the surface, at the bottom

L92: remove "formation"

L98: to the hypothesis that hypoxic

L100: Zhang et al. (2018) recently demonstrated that; and, as a result, location

L101: are variable

L102: do not

L104: in a more general

L106/107: remove "It is clear"; result in higher DO consumption rates

L114/115: in the observed spatial patterns of temperature, salinity, chlorophyll *a* (Chl *a*) and DO

L128: inside the river estuary

L129: At all stations, vertical

L134: cruise, Chl *a*

L140: state that Chl *a* was not sampled in 2017, which is why satellite-derived Chl *a* is used

L143-145: remove this sentence and only state how sigma was calculated in the figure caption where sigma is used (Fig. 6a, b).

L147: for the estimation of the spatial extent of CDW

L148: remove "well"

L151: Could you provide a link to the Copernicus Marine Service website?

L152: Large and Pond (1981).

L153: What are rho_air and U?

L164: during the reference period

L177: Avoid using UBD. Instead refer to the isoline in the text later on

L178: add references for the different criteria; originates

L179: Avoid using UBK. Instead refer to the isoline in the text later on

L182: in Xu et al. (2018); used in this study

L183: The width

L186-187: define "g" and the alpha in the equation looks odd

L192: no abbreviation in title

L193: observations, we analyse

L194: distributions observed in the summers of 2015 and 2017

L196: Here you discuss the 25 (psu) isoline, but in the figures you show the 30 isoline. Please show the one you need for the description

L198: In 2017, the

L199: Water fresher than 25 was …

L202: The sea surface temperature; in 2015 (Fig. 3c) than in 2017 (Fig. 3d).

L204: remove "(Fig. 3c-d)"

L212/213: The calculation of the density difference is bottom minus surface value.

L214: in the north

L215: south of the

L216: and a cold and salty

L218: (Fig. 6a) as the dense bottom layer water

L222: of the coastal slope; Downwelling had pushed

L223: Probably, onshore transport

L311: causing the current

L314: in the further discussion

L315: Comparing the two years' discharge

L319: remove "we"; remove "(wind forcing)"

L322: According to Eq. (4), the width …

L324: 55 km for the inflow

L325: remove "were considered"; "partly" instead of "somewhat"; explains

L329: could have caused the differences between

L335: As described previously

L336: before the sampling

L339: toward

L340: did not prevail

L342: "Probably" instead of "Likely"

L348: Maybe "subsurface" instead of "deep layer"?

L353: This is the combined

L355: such phenomenon did not occur/prevail

L358: An opposed gradient; coastline, was

L362: currents toward

L365: You write "First", but there's no "Second"

L370: correlates well

L379: occur on shorter time scales, probably cause

L381/382: move "also" after "likely"

L384: remove "well"; flatter

L385-391: remove sentences about eddy, it's too speculative and distracts from the main story

L394: We further analysed

L396: and a northward; should it be "t_c>>0" to distinguish from weak northward winds on L398?

L402: The CDW was mainly transported

L405: flow

L407: include reference to figure (panels) that show this northeastward spreading

L425: explains why

L426: is required

L430: refer chronologically to panels of Fig. 12 (currently a, c are referred to after b, d), maybe just rearrange the panels accordingly

L442: accompanied by; the spreading; nutrient-rich

L449: remove the text in the parentheses, it's a bit confusing

L451: Southeasterly winds and higher discharge during summer monsoon both favor …

L454: late August/early September; and are reflected in

L455/456: only use "On the other hand" if you used "On the one hand" in an opposing statement just before; Wind speed and discharge were close to average during summer 2017, while wind speed was much weaker in 2015

L457: remove "concurrently"

L462: remove "paper"

L463: also related; the corresponding surface

L465: The monthly occurrence […] September 2018 (Fig. 11) clearly shows that CDW …

L467: one can see that 2015 differs with low values

L468: maps

L469: do not; in the north

L470: compared to; occurred in the south in October; include a reference for this deterioration; This demonstrates that the wind

L474: close to the long-term; will more likely occur east of the

L478: than the long-term

L480: "extent" instead of "spreading area"

L492: 2018, stratification

L502: and, according to the latter factors, estimate

L506: fate; separated by regions; What do you mean with bulge?

L507: downstream the coastal current

L509: of the coastal current

L519/520: remove statement on eddy

L524: was also observed by Yang et al.

L528: nitrogen-to-phosphorus (N and P have not been introduced)

L536: by a typhoon

L539: a valuable time series of DO in the near-bottom layer

L543: of the KSSW as DO declined

L550: The importance

L551: is also revealed when near-bottom; maps of vertically integrated AOU

L552: compared to

L572: holds according to our analyses

L577: In the latter area, KSSW was present but there …

L584: effect of the thermocline

L585: add reference for previous oxygen depletion of subsurface water

L588: compared to

L590: is an important

L591: lateral DO supply

L592: nutrient-rich

L593: sinking causes DO consumption

L598: only use "On the other hand" if you used "On the one hand" in an opposing statement just before

L601: not as pronounced

L602: The continuous; was demonstrated by; "monthly" is not really " continuous"

L604: Second

L605: on its way south

L606: compared to the northern region

L607: "penetrate" or "diminish" instead of "destroy"?

L608: remove "In short"

L617: by KSSW intrusion or CDW spreading

L619: below the pycnocline

L620: Both distribution of CDW and KSSW and the occurrence of KSSW upwelling are controlled by the wind field.

L621: alters the Chinese Coastal Current by creating an Ekman surface flow

L623: on the coastal slope, resulting in upwelling
L625: In consequence, the
L633: remove sentence starting with "Concepts"
L634/635: Our results further suggest that a combination of wind field data, remotely sensed sea-surface salinity and sea level could be used to forecast the hydrographic conditions and potential hypoxic area prior to field work. (NOTE: You don't really need river discharge if you have sea surface salinity information.)
L648: led
L647: during the 2015 cruise

Fig. 1: add "depth (m)" label to color scale; the cross with the wind location, the large circle with the mooring and the sections N15/N17 are barely visible in gray-scale
Fig. 3: Is it on purpose that excluded stations (in the estuary) are shown on the maps for 2017 but not for 2015? Change order of isolines in caption as you first show salinity
Figs. 4/5: Maybe change order of first and second row as you first describe S and then T in the manuscript. I suggest adding a few more increments to the color scales (e.g. 20, 21, 22, 23, 24, 25, 26 for temperature), perhaps you can then match the number of colors in the panels to the increments (then you could also remove the green lines, which I find a bit confusing). Maybe use longitude for the x axes to allow for easier match-up with the locations in Fig. 1. I am a bit confused by the fact that the number of station locations (small crosses) in the first row doesn't match the number of stations along each transect in Fig. 1
Fig. 8: The red line for the current measurements is barely visible in gray-scale. Positive and negative values in the last two panels are indistinguishable in gray-scale. Add units to the color scales. I suggest removing the longitude/latitude information for all panels except the bottom-left to save some space between panels (that would allow for larger panels). Use white space between "cm" and "s" in the units in the caption.
Fig. 9: at section S1 (see Fig. 8); use white space between "m" and "s" in units of axes
Fig. 11: I suggest removing the longitude/latitude information for all panels except the bottom-left to save some space between panels (that would allow for larger panels). Why do you use g/kg as salinity unit in the caption? In the methods you wrote it's practical salinity scale. Better be consistent. If the satellite product is in g/kg, you should convert to PSU..

Table 1: I don't find this table very useful. I think you could remove it.

---

## Referee Report (RR2)

**Review of Liblik et al.: "Wind-driven stratification patterns and dissolved oxygen depletion off the Changjiang (Yangtze) Estuary"**

The authors have satisfactorily addressed my comments and/or responded in the letter of response. I am generally satisfied with this new version of the manuscript and I only have one minor comment and a few technical suggestions/corrections.

Therefore, I recommend publication after very minor revisions/technical corrections. My specific comments follow below.

**Specific comment**

Zheng et al. (2016; DOI: 10.5194/nhess-16-2559-2016) published a detailed modelling study on the effect of the wind field (and Changjiang River discharge) on oxygen in the East China Sea. It would be nice if the authors could put their results in relation to the work by Zheng et al. as part of the Discussion.

**Technical corrections**

Line 16: "freshwater"

Line 17: "Seasonal hypoxia"

Line 27: "of the Chinese"

Line 28: "observed in the summer of 2015"

Lines 29-32: I suggest to change order of these two sentences and slightly rephrase as the offshore Ekman transport causes the upwelling (i.e. state the cause first, effect second).

Lines 33/34: Isn't deep water intrusion and coastal upwelling the same?

Line 35: "in the physical environment"

Line 55: "phenomenon in the East China Sea (ECS) off the Changjiang Estuary"

Line 56: "has been expanding"

Line 58: "when" instead of "as the"

Lines 59/60: Please state the most important processes, otherwise it's too generic.

Line 61: "into the areas north and south of 30°N"

Lines 67-69: You could consider citing Große et al. (2019) here (as you do on line 102). We quantify the influence of Changjiang nutrients on oxygen consumption in the region and, thus, explicitly show this relation.

Lines 75-77: I think you could remove this part on tides here as tides are not really part of your analyses. Instead, you could consider including the reference in the Discussion where you discuss the potential use of gliders and their potential limitations due to strong tides (lines 528/529).

Line 88: "in the ECS" (and introduce 'ECS' on line 55 as suggested)

Line 92: Not clear what this "cumulative effect" is meant to be.

Line 104: "favourable conditions for the formation and maintenance of hypoxia" (remove the rest after "for")

Line 109: "during two cruises"

Line 110: "links:

Lines 121/122: move "respectively" to end of sentence

Line 130: "before they have been preserved"

Line 136: "is given"

Line 148: Here and in all following explanations of equations, make sure that variables are type-set the same way as in the equations, i.e. cursive letters also in the text.
Line 152: "(1/12°) resolution and standard vertical levels with resolution increasing"
Line 155: "finer-scale"
Line 165: "calculated as follows"
Equation (3): suggest to write "AOU" in non-cursive letters (or cursive everywhere in the text)
Line 172: "the oxycline"
Line 175: "Therefore, we"
Line 203: "water depth"
Line 205: "water depth"
Line 208: "are described"
Line 213: "From there, stratification declines rapidly"
Line 215: "offshore and Fig." This seems incomplete?
Line 216: remove "distance"; add "(indicating upwelling)" after "offshore"
Line 224: no white space after "<"
Line 225: "where the cold"; "area, which was"
Line 226: "but where bottom"; "The stratification pattern"
Lines 228/229: move "respectively" to end of sentence
Lines 231/232: Please state explicitly beforehand what you mean with this "coupling"; "riverine freshwater"
Line 235: "illustrate"
Line 237: "pushed deeper"
Line 245: "by high"
Line 246: remove one dash between "3-5"; "rapidly onshore"
Line 247: remove one dash between "2-3"; "by the cold"
Line 251: "We, therefore, hypothesize"
Line 254: "low-salinity"
Line 267: remove "enough"
Line 269: no "-" after "temperature" and "salinity"; no comma after "salinity"
Line 280: use "KSSW" instead of "Kuroshio water"?
Line 288: "patterns? What"
Line 301: white space between "m3" and "s-1"
Line 298: white space between "m3" and "s-1"; "one week" instead of "1 week"
Lines 304-306: First, you write "accumulated run-off" (i.e. temporally integrated), then "Mean river flow" and you use mean flow later. Please be consistent; white space between "m3" and "s-1"; "the plume"
Line 307: "According to Eq. (4)"
Line 314: add reference to corresponding Fig. after "pycnocline"
Line 318: "wind forcing"
Line 333: "wind caused"
Line 334: "suggests"
Line 335: What do you mean with "pattern of processes"? You don't show processes. Please rephrase.
Line 342: "simulated current field"
Line 367: "probably cause"
Line 372: "Seafloor"
Line 381: "as salinity < 30"

Line 387: "describes well"

Lines 389/390: This statement doesn't add much unless you conclude something from it. I'd remove it.

Line 394: "we attempt"; comma after "velocities"

Line 421: "hypoxia in the ECS."

Line 425: comma instead of dash after "oxygen"

Line 429: add reference to corresponding figure (panel) at and of sentence

Line 430: add reference to corresponding figure (panel) at and of sentence

Line 431: "were observed in August and October 2006"; Do you mean similar to the 2015 patterns? If so, move this sentence one sentence up.

Line 438: "in DO and"

Lines 442/443: (Fig. 3 in Wang et al., 2012)"

Line 444: add reference to Fig. 5

Lines 451/452: Rephrase this sentence such that it "stands out in terms of river discharge"

Line 463: remove "distribution"

Line 466: "has been reported"

Line 467: has been registered in the north"

Line 472: "lower, considerable DO"

Line 473: "on top"

Line 475: "are likely to influence"

Line 477: "as well as"

Line 480: "separated into"; "the circulating bulge"

Line 481: "the downstream coastal current

Line 482: "discharge is transported alongshore with the coastal"

Lines 486/487: move "in winter" to end of sentence

Line 490: "physical and chemical"; "including DO conditions"

Line 493: no white spaces between "upwelling" and "CDW"; I still find the term "coupling" confusing. What about "interaction"?

Line 497: "are considerably"

Line 498: "and, therefore, promote"

Line 517: comma after "water"; "), rather than"

Line 520: "DO maps"

Line 522: "DO-depleted"

Line 524: "Studies of pycnocline dynamics"

Lines 527/528: "in the necessary spatio-temporal resolution"

Line 536: "for hypoxia formation"

Line 545: "The rest of"

Line 546: no comma after "present"

Line 553: If you use "First", you need to use "Second" at a later point. Maybe remove "First"?

Line 554: "DO consumption"

Line 555: " The stations in the farthest southeast"

Line 557: "was still"

Lines 558/559: "DO depletion"; "DO transport"

Line 568: "section N15"

Line 569: "can disappear"

Line 587: "The presence of a pycnocline"

Line 592: "the Chines"

Line 595: Rephrase so that it is clear that "shallow" means "up to close to the surface" and not in shallow areas
Line 596: "the Chinese"
Line 598: "contributions to"
Line 600: "the rest"
Line 602: "sensitivity of the boundaries"; move "considerably" to end of sentence
Line 619: "analyses:
Fig. 7: Don't use "UBD" and "UBK" in caption
Fig. 8: depth is not negative. It is negative z but depth itself is positive. Please change color scale labels.

---

## Author Response (AR3)

Dear Dr. Fennel and reviewers

Please find the revised manuscript and responses to the comments. Thank you again for taking the time with our paper.

Best Regards,
Taavi Liblik

**Responses to reviewer 1**

Comment: Zheng et al. (2016; DOI: 10.5194/nhess-16-2559-2016) published a detailed modelling study on the effect of the wind field (and Changjiang River discharge) on oxygen in the East China Sea. It would be nice if the authors could put their results in relation to the work by Zheng et al. as part of the Discussion.

Reply: Thank you for that recommendation. Yes, we should mention the paper in our study.

Action: We mention the study now around line 445 (concerning hypoxic area sensitivity to wind direction), line 467 (river discharge vs. hypoxic area), 507 (wind speed vs. hypoxia).

**Technical corrections**

Thank you again for your corrections! We have modified the manuscript according to your comments. The comments that needed a reply are given below.

Lines 33/34: Isn't deep water intrusion and coastal upwelling the same?

Reply: No, it is not the same. By upwelling we mean the surfaced former subsurface water. Deep water intrusion stands for deep water advected to the area (in our case to the north), but it could be not surfaced.

Action: no action.

Lines 75-77: I think you could remove this part on tides here as tides are not really part of your analyses. Instead, you could consider including the reference in the Discussion where you discuss the potential use of gliders and their potential limitations due to strong tides (lines 528/529).

Reply: We think it is important to shortly highlight (it is two lines only) the importance of wind compared to tides on stratification in the introduction. Yes, we can add a bit more on gliders.

Action: We have added to the discussion: „Despite strong tidal oscillations the vertical average of residual current speeds in the area (Peng et al., 2017) is smaller than the forward speed of gliders. Moreover, adaptive piloting methods can be used to control a glider (Chang et al., 2015)."

**Responses to reviewer 2**

Comment: The abstract needs improvement, a more concise description of processes in these two years and a more specific conclusion about interannual variability. This study focuses on mechanisms on "The seasonal hypoxia in the area has variable location and range" and also interannual variability. So, "the decadal trend reveals expansion and intensification of the dissolved oxygen (DO) depletion" is not necessary here. Observation months should be given in the abstract to make it clear that observations are in different months. And these observations can reflect seasonal variations, as line437-438 says. The conclusion in the last sentence seems too general. The statistical analysis on causes of interannual variability (the relationship between winds and CDW and hypoxia distribution) is of more scientific value. And the finding about it should be mentioned in the abstract.

Reply: Thank you for the suggestions! We agree some changes can be done to highlight the main results better.

Action: We removed the statement about the trend and added observation months. We changed the abstract to highlight the results about inter-annual variability better.

Comment: The conclusions just highlight processes controlling on seasonal variation. The controlling processes associated with interannual variability should be also mentioned. The authors are also expected to improve the language.

Reply: Yes, we agree. Inter-annual variability should be mentioned there.

Action: We have added text about interannual variability to the conclusions and we have made another language check.

Specific comments:

Line29: The value of "hypoxic area" is in 2015, how about 2017?

Reply: We did not capture the northern border of the hypoxic area in 2017. The estimation would be too speculative.

Action: no action.

Line50: I suggest defining hypoxia in the Introduction to make it clear for readers.

Reply: Yes, we can do it.

Action: We have added: "Hypoxia is a condition of low DO, which cannot be sustained by marine life. Depending on marine organisms, hypoxia can have various definitions (Vaquer-

Sunyer and Duarte, 2008). In the present study, hypoxia is defined as a DO concentration of $<3.0$ mg $l^{-1}$."

We removed the last two sentences from "data and methods" part to avoid repeating.

Line209: Stratification intensity (for example buoyancy frequency, N2) can be calculated, because both temperature and salinity are available. It would be clearer to indicate weak or strong stratification than surface-bottom density difference.

Reply: Thank you for the suggestion. Yes, it can be interesting. This calculation (inc. buoyancy frequency calculation) was done in the preparatory phase of the paper, but we left it out from the manuscript. Yes, it can show the strength of stratification locally, but the gradients here are so strong that oxygen depletion (which is the main focus of the paper) linkage to the thermocline is obvious (Fig. 4-5). We think Fig. 6 (density difference) together with the upper boundaries of the KSSW and DO depletion (Fig. 7) give a good overview of the depth and strength of the stratification and their linkage to the oxygen depletion. As we have argued at lines 539-552 strength of the stratification itself does not necessarily reflect in the intense oxygen depletion. It is rather the presence of the KSSW and CDW and/or upwelling at the same time, which leads to hypoxia. We are afraid adding another stratification characteristics does not give extra value to the main storyline of this paper. Chapter 3.1. was too long before the last revision (rised by reviewer 1) and we do not want to make it longer again.

Action: no action.

Line272: I would also suggest that the thermocline/pycnocline intensity and position can be calculated here. There is criterion about thermocline/pycnocline for shelf waters.

Reply: This was also done in the preparatory phase of the paper. We also calculated the contribution of the temperature and salinity to the stratification strength. However, when the final version of the submitted paper was ready, we realized this information does not contribute much to the conclusions of the paper. We think more detailed characteristics of stratification concerning oxygen depletion in the area is important to analyze, but this work must be done in future studies. We think an intense measurement campaign with autonomous devices (profilers, gliders) would allow us to capture evolution in stratification parameters in necessary spatial and temporal scale. We are afraid there is already enough information about stratification under chapter 3.1.

Action: no action.

Line440: Wang (2012) also mentioned the influence of Kuroshio intrusion with a statistical analysis.

Reply: Yes, we agree

Action: We complemented the sentence accordingly.

Line440-445: I do not think the relation to seasonality of wind, CDW and Kuroshio intrusion is your new finding.

Reply: In the discussion chapter we discuss our findings concerning earlier studies. We think the role of wind on advection (of CDW and KSSW) is important and this is not highlighted by Wang et al. (2012) study. However, we decide to remove these sentences. The main argument for doing that was the fact that we did not analyze wind data in detail for the year (2006) studied by Wang et al. (2012).

Action: This part was removed.

Line453-454: What does "the latter" mean? Not quite clear here. Here, the conclusion is just proved by the year of 2017.

Reply: We have rephrased "the latter" to avoid confusion. Yes, here we cite to year 2017, that is why we use here wording "indicates". However, this conclusion is supported by the following discussion, e.g. remote sensing observations in 2016 (Fig. 11), in-situ observations by Zhou et al. 2019 and wind statistics (Fig. 12).

Action: It says now: Thus, it indicates if wind forcing is close to the long-term average, hypoxia will more likely occur east of the river mouth (northern part of the study area) and hypoxia occurrence in the south is more limited."

Line521-523: It is confused here. Hypoxia is severer in north part in 2017 that in the south part in 2015. And the thickness of hypoxic waters in these two year looks similar, about 20-30 m thick. But it deduced that there is a thicker oxygen-depleted layer in the south part in 2015 according to integrated AOU. The authors can calculate hypoxic water thickness to quantify it.

Reply: We cited the thickness of the oxygen-depleted layer, not to the hypoxic layer. The thickness of the layer was 38-40 m in 2015 and 27-28 m in 2017.

Action: We added this information about thicknesses to the results subchapter 3.1.

Line585-586: The two main conditions here, enhanced primary production and KSSW intrusion, are not coordinate factors. Enhanced primary production can be caused by KSSW intrusion induced upwelling. Consider "enhanced primary production and pycnocline dominated by KSSW intrusion and CDW"?

Reply: Yes, we understand what you mean and we considered your suggestion. We have explained the same a bit longer in the following two sentences. We think it is better to keep the longer version for a reader, who is not familiar with the processes here.

Action: no action

Line49: A comma should be here instead of a dot. It should be a sentence not two sentences.

Reply: We cannot see the advantage of having a comma instead of a dot at line 49.

Action: no action

Line238: at one station

Reply and action: done.

[revised manuscript text omitted]